# Quantification and discovery of sequence determinants of protein-per-mRNA amount in 29 human tissues

Basak Eraslan[1,2,†], Dongxue Wang[3,†] (iD), Mirjana Gusic[4,5], Holger Prokisch[4,5] (iD), Björn M Hallström[6], Mathias Uhlén[6] (iD), Anna Asplund[7], Frederik Pontén[7] (iD), Thomas Wieland[3], Thomas Hopf[3], Hannes Hahne[8,*] (iD), Bernhard Kuster[3,9,**] (iD) & Julien Gagneur[1,***] (iD)

## Abstract

Despite their importance in determining protein abundance, a comprehensive catalogue of sequence features controlling protein-to-mRNA (PTR) ratios and a quantification of their effects are still lacking. Here, we quantified PTR ratios for 11,575 proteins across 29 human tissues using matched transcriptomes and proteomes. We estimated by regression the contribution of known sequence determinants of protein synthesis and degradation in addition to 45 mRNA and 3 protein sequence motifs that we found by association testing. While PTR ratios span more than 2 orders of magnitude, our integrative model predicts PTR ratios at a median precision of 3.2-fold. A reporter assay provided functional support for two novel UTR motifs, and an immobilized mRNA affinity competition-binding assay identified motif-specific bound proteins for one motif. Moreover, our integrative model led to a new metric of codon optimality that captures the effects of codon frequency on protein synthesis and degradation. Altogether, this study shows that a large fraction of PTR ratio variation in human tissues can be predicted from sequence, and it identifies many new candidate post-transcriptional regulatory elements.

**Keywords** codon usage; mRNA sequence motifs; proteomics; transcriptomics; translational control

**Subject Categories** Genome-Scale & Integrative Biology; Methods & Resources; RNA Biology
**Mol Syst Biol. (2019) 15: e8513**

## Introduction

Unraveling how gene regulation is encoded in genomes is central to delineating gene regulatory programs and to understanding predispositions to diseases. Although transcript abundance is a major determinant of protein abundance, substantial deviations between mRNA and protein levels of gene expression exist (Liu *et al*, 2016). These deviations include a much larger dynamic range of protein abundances (García-Martínez *et al*, 2007; Lackner *et al*, 2007; Schwanhäusser *et al*, 2011; Wilhelm *et al*, 2014; Csárdi *et al*, 2015) and poor mRNA–protein correlations for important gene classes across cell types and tissues (Fortelny *et al*, 2017; Franks *et al*, 2017). Moreover, deviations between mRNA and protein abundances are emphasized in non-steady-state conditions driven by gene-specific protein synthesis and degradation rates (Peshkin *et al*, 2015; Jovanovic *et al*, 2016). Therefore, it is important to consider regulatory elements determining the number of protein molecules per mRNA molecule when studying the gene regulatory code.

Decades of single-gene studies have revealed numerous sequence elements affecting initiation, elongation, and termination of translation as well as protein degradation. Eukaryotic translation is canonically initiated after the ribosome, which is scanning the 5′ UTR from the 5′ cap, recognizes a start codon. Start codons and secondary structures in 5′ UTR can interfere with ribosome scanning (Kozak, 1984; Kudla *et al*, 2009). Also, the sequence context of the start codon plays a major role in start codon recognition (Kozak, 1986).

1 Computational Biology, Department of Informatics, Technical University of Munich, Garching, Munich, Germany
2 Graduate School of Quantitative Biosciences (QBM), Ludwig-Maximilians-Universität München, Munich, Germany
3 Chair of Proteomics and Bioanalytics, Technical University of Munich, Freising, Germany
4 Institute of Human Genetics, Technical University of Munich, Munich, Germany
5 Institute of Human Genetics, Helmholtz Zentrum München, Neuherberg, Germany
6 Science for Life Laboratory, KTH - Royal Institute of Technology, Stockholm, Sweden
7 Department of Immunology, Genetics and Pathology, Science for Life Laboratory, Uppsala University, Uppsala, Sweden
8 OmicScouts GmbH, Freising, Germany
9 Center For Integrated Protein Science Munich (CIPSM), Munich, Germany
*Corresponding author. Tel: +49 8161 9762892; E-mail: hannes.hahne@omicscouts.com
**Corresponding author. Tel: +49 8161 71 5696; E-mail: kuster@tum.de
***Corresponding author. Tel: +49 89 289 19411; E-mail: gagneur@in.tum.de
†These authors contributed equally to this work

The translation elongation rate is determined by the rate of decoding each codon of the coding sequence (Sorensen *et al*, 1989; Gardin *et al*, 2014; Hanson & Coller, 2018). It is understood that the low abundance of some tRNAs leads to longer decoding time of their cognate codons (Varenne *et al*, 1984), which in turn can lead to repressed translation initiation consistent with a ribosome traffic jam model (reviewed in Hanson & Coller, 2018). However, estimates of codon decoding times in human cells and their overall importance for determining human protein levels are highly debated (Plotkin & Kudla, 2011; Quax *et al*, 2015; Hanson & Coller, 2018). Secondary structure of the coding sequence and chemical properties of the nascent peptide chain can further modulate elongation rates (Qu *et al*, 2011; Artieri & Fraser, 2014; Sabi & Tuller, 2017; Dao Duc & Song, 2018). Translation termination is triggered by the recognition of the stop codon. The sequence context of the stop codon can modulate its recognition, whereby non-favorable sequences can lead to translational read-through (Bonetti *et al*, 1995; McCaughan *et al*, 1995; Poole *et al*, 1995; Tate *et al*, 1996). Furthermore, numerous RNA binding proteins (RBPs) and microRNAs (miRNAs) can be recruited to mRNAs by binding to sequence-specific binding sites and can further regulate various steps of translation (Baek *et al*, 2008; Selbach *et al*, 2008; Guo *et al*, 2010; Gerstberger *et al*, 2014; Hudson & Ortlund, 2014; Cottrell *et al*, 2017). However, not only predicting the binding of miRNAs and RBPs from sequence is still difficult, but the role of few of these binding events in translation is well understood.

Complementary to translation, protein degradation also plays an important role in determining protein abundance. Degrons are protein degradation signals which can be acquired or are inherent to protein sequences (Geffen *et al*, 2016). The first discovered degron inherent to protein sequence was the N-terminal amino acid (Bachmair *et al*, 1986). However, the exact mechanism and its importance are still debated, with recent data in yeast indicating a more general role of hydrophobicity of the N-terminal region on protein stability (Kats *et al*, 2018). Further protein-encoded degrons include several linear and structural protein motifs (Ravid & Hochstrasser, 2008; Geffen *et al*, 2016; Maurer *et al*, 2016), or phosphorylated motifs that are recognized by ubiquitin ligases (Mészáros *et al*, 2017). Altogether, numerous mRNA and protein-encoded sequence features contribute to determining how many protein molecules per mRNA molecule cells produce. However, it is known neither how comprehensive the catalogue of these sequence features is nor how they quantitatively contribute to protein-per-mRNA abundances.

To address these questions in a human cell line, Vogel and colleagues (Vogel *et al*, 2010) performed multivariate regression analysis to predict protein abundances from mRNA abundances and mRNA sequence features. This seminal work was based on transcriptome and proteome data for a single cell type, Daoy medulloblastoma cells. Whether the conclusions drawn at the time can be generalized genome-wide and to other human cell types remains an open question. Moreover, transcriptomics and proteomics technologies at the time were not as sensitive and quantitative as they are today, leaving reliable quantification only for 476 protein-coding genes for further analysis. These 476 proteins were among the most abundant proteins, therefore leading to possibly strong analysis biases. Furthermore, this study focused on known sequence determinants of protein-per-mRNA abundances and refrained from discovering novel sequence elements.

Here, we exploited matched proteome and transcriptome expression levels for 11,575 genes across 29 human tissues (Fig 1A, Wang *et al*, 2019) to predict protein-to-mRNA ratios (PTR ratios) from sequence. To interpret our findings related to mRNA degradation (Radhakrishnan & Green, 2016), translation, and protein degradation, we included mRNA half-life measurements (Tani *et al*, 2012; Schueler *et al*, 2014; Schwalb *et al*, 2016), in addition to human ribosome profiling of 17 independent studies (Dana & Tuller, 2015; O'Connor *et al*, 2016) as well as protein half-life measurements from immortal and primary cell lines (Zecha *et al*, 2018; Mathieson *et al*, 2018; Fig 1A). We considered known post-transcriptional regulatory elements and identified novel candidates in the 5′ UTR, coding sequence, and 3′ UTR, by means of systematic association testing. We also modeled the effect of codons on protein-to-mRNA ratio, leading to a new quantitative measure of codon optimality which we compared to existing metrics. Our integrative model estimates the contribution of all these elements on protein-to-mRNA ratio and predicts tissue-specific PTR ratios of individual genes at a relative median error of 3.2-fold. Finally, we are providing initial experimental results to assess the functional relevance of the novel potentially regulatory elements.

# Results

## Matched transcriptomic and proteomic analysis of 29 human tissues

Using label-free quantitative proteomics and RNA-Seq, we profiled the proteomes and transcriptomes of adjacent cryo-sections of 29 histologically healthy tissue specimens collected by the Human Protein Atlas project (Fagerberg *et al*, 2014) that represent major human tissues (Wang *et al*, 2019). To facilitate data analysis, we modeled every gene with a single transcript isoform because there was little evidence for widespread expression of multiple isoforms and to avoid practical difficulties of calling and quantifying isoform abundance consistently at mRNA and protein levels. The number of genes with multiple quantified isoforms on protein level was small (10% of the 13,664 genes with a protein detected in at least in one tissue). Also, for 5,636 (43%) genes the same isoform was the most abundant one across all tissues (out of 12,978 genes with at least one mRNA transcript isoform expressed [FPKM > 1] in at least in one tissue; Materials and Methods, Appendix Fig S1). Moreover, 4,303 (34%) genes had a perfect match between the RNA-Seq-defined and the proteomics-defined major isoform in all the tissues they were detected (out of 12,920 genes with matched protein and mRNA measurements). For the remaining genes, there were some mismatches between the RNA-Seq-defined and the proteomics-defined major isoforms in a varying number of tissues, yet the number of matched RNA-Seq-defined and proteomics-defined major isoforms were larger than the unmatched ones in almost all tissues (Appendix Fig S2). Since we were restricted by the small number of isoform counts on proteome level, we defined the transcript isoform with the largest average protein abundance across tissues as its major transcript isoform. We used these major transcript isoforms to compute all sequence features and mRNA levels for all tissues (Materials and Methods). The mRNA levels were estimated from RNA-Seq data by subtracting length and

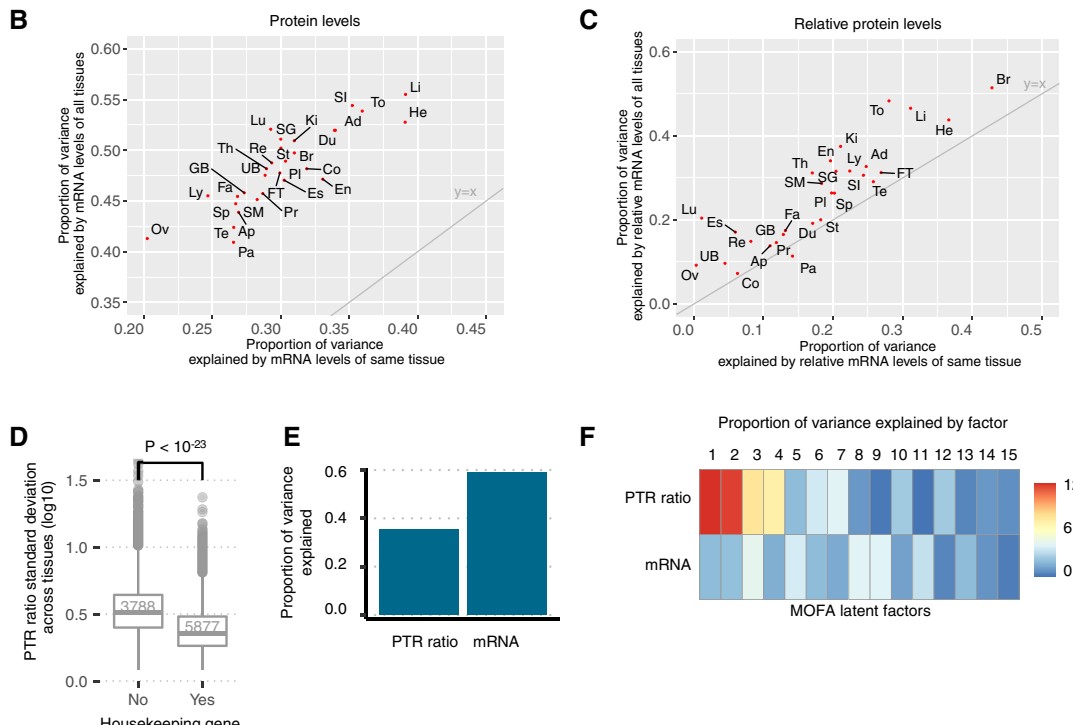

**Figure 1.  Variation of protein and mRNA levels across 29 human tissues.**

A   Overview of the datasets analyzed in this study. We analyzed the protein-to-mRNA ratios by considering a dataset of matched proteome and transcriptome of 29 human tissues (Wang *et al*, 2019). We further interpreted our findings with respect to ribosome occupancy datasets, reflecting translation elongation, protein half-life datasets, and mRNA half-life datasets. Solid lines represent the dependencies in the basic gene expression kinetic model. Dashed line represents the coupling between mRNA elongation and degradation rates (Radhakrishnan & Green, 2016).

B   Proportion of the variance (Materials and Methods) explained by mRNA levels of all tissues (*y*-axis) against proportion of the variance explained by mRNA levels of the same tissue (*x*-axis) of protein levels for 29 human tissues. The gray line is the identity line *y = x*. Ad (Adrenal), Ap (Appendices), Br (Brain), Co (Colon), Du (Duodenum), En (Endometrium), Es (Esophagus), FT (Fallopian tube), Fa (Fat), GB (Gall bladder), He (Heart), Ki (Kidney), Li (Liver), Lu (Lung), Ly (Lymphnode), Ov (Ovary), Pa (Pancreas), Pl (Placenta), Pr (Prostate), Re (Rectum), SG (Salivary gland), SI (Small intestine), SM (Smooth muscle), Sp (Spleen), St (Stomach), Te (Testis), Th (Thyroid), To (Tonsil), UB (Urinary bladder).

C   Same as in (B) for relative levels, i.e., the log-ratios of the levels with respect to the median level across tissues. The gray line is the identity line *y = x*.

D   Distribution of the standard deviation across tissues of the PTR ratio (log₁₀) for housekeeping genes (left) and other genes (right). The PTR ratio for housekeeping genes varies significantly less than for other genes (Wilcoxon test). Shown are the quartiles (boxes and horizontal lines) and furthest data points still within 1.5 times the interquartile range of the lower and upper quartiles (whiskers).

E   Proportion of variance across tissues of PTR ratio (left) and mRNA (right) explained by the 15 latent factors fitted by joint optimization of the likelihood of both data modalities (Argelaguet *et al*, 2018).

F   Proportion of variance in mRNA and PTR ratio across tissues explained by each latent factor fitted by Multi-Omics Factor Analysis (MOFA) (Argelaguet *et al*, 2018). Factors that are active in both mRNA and PTR ratio capture shared covariation across tissues, and factors that are active in only one capture the signal specific to that modality.

sequencing-depth-normalized intronic from exonic coverages (Materials and Methods). Subtracting intronic coverage led to slightly improved correlations between mRNA and protein levels in every sample (Appendix Fig S3), possibly because it better reflects the concentration of mature mRNAs, which are the ones exposed to the translation machinery. RNA-Seq technical replicates were summarized using the median value. Requiring at least 10 sequencing-depth-normalized reads per kilobase pair further improved the correlation between mRNA and proteins, likely because low expression values on transcript and protein levels are associated with a larger measurement error. Lastly, we restricted the analysis to transcripts with a 5′ UTR and 3′ UTR longer than 6 nt to make sure that all considered sequence features could be computed. Altogether, this analysis led to matched quantifications of protein and mRNA abundances for 11,575 genes across 29 tissues (Tables EV1–EV4), where an average of 7,972 (69%, minimum 7,300 and maximum 8,869) PTR ratios were quantified per tissue.

## Protein-to-mRNA ratio variation across genes in each tissue

How well mRNA levels explain protein levels and the importance of post-transcriptional regulation in adjusting tissue-specific proteomes has been highly debated over the last 10 years (Maier et al, 2009; Lundberg et al, 2010; Schwanhäusser et al, 2011; Li et al, 2014; Wilhelm et al, 2014; Csárdi et al, 2015; Edfors et al, 2016; Fortelny et al, 2017; Franks et al, 2017). In every tissue, the proportion of variance of protein levels across genes explained by mRNA levels of the same tissue (Fig 1B, x-axis, Materials and Methods) ranged from 20% (ovary) to 39% (liver). However, we observed that much larger proportions of the variance could be explained by using mRNA profiles across all tissues (between 41% for pancreas and 56% for liver, Fig 1B, y-axis, Materials and Methods, $P < 10^{-132}$ for each tissue). The reasons for this increase in explained variance are at least twofold. Biologically, it is conceivable that co-expression patterns of mRNAs can be predictive for post-transcriptional regulation because functionally related genes are co-regulated at the mRNA level and at the post-transcriptional level (Franks et al, 2017). Technically, this increase may also be driven by the more robust nature of mRNA profiles across all tissues compared to mRNA level measures in a single tissue. This is consistent with observations by Csárdi et al (2015) that de-noising of mRNA measurements of budding yeast can enhance the explained variance of protein levels.

## Protein-to-mRNA ratio variation of genes across tissues

Variation of the PTR ratio per gene across different tissues is more relevant for understanding the tissue-specific post-transcriptional regulation of protein expression than the variation between different genes of a single tissue. Our analysis shows that the variation of the PTR ratio of single genes across tissues was small in comparison with the variation of PTR ratios across different genes (Fig EV1A and B). To study the variations per gene across tissues, we defined the relative protein level as the log-ratio of the protein level compared to its median across tissues. We similarly defined the relative mRNA level. The relative mRNA levels of the same tissue explained only between 0% (ovary) and 43% (brain) of the relative protein level variance suggesting that tissue-specific PTR regulation plays an important role in determining tissue-specific protein levels (Fig 1C). These two

observations are consistent with earlier analyses which were also performed across human tissues (Franks et al, 2017). More interestingly, we found that between 7% (colon) and 51% (brain) of the variance in relative protein levels could be explained when considering the relative mRNA levels of all tissues (Fig 1C, Materials and Methods, every tissue with a significant increase $P < 10^{-19}$, except for pancreas), which again indicates that co-expression patterns of mRNAs may be predictive of post-transcriptional regulation. Evidence for co-regulation of PTR ratio was corroborated by gene set enrichment analyses. Among the considered genes, housekeeping genes defined by the Human Protein Atlas, which are abundantly expressed in general, had fairly similar PTR ratios across tissues (Fig 1D). Gene set enrichment analysis (FDR < 0.1) performed with DAVID (Huang et al, 2009a,b) revealed that cellular protein complex assembly, negative regulation of protein metabolic process, and regulation of cytoplasmic transport were some of the biological processes enriched for genes with low PTR ratio standard deviation (Fig EV1C). Also, proteins localized in certain cellular components such as chaperonin-containing T-complex, whole membrane, and cytoskeleton had significantly low PTR ratio standard deviation across tissues (Fig EV1D). In contrast, genes with strongly varying PTR ratios across tissues were enriched in biological processes that point toward tissue-specific and cell-specific biology and include cilium organization, glycolipid biosynthetic process, single–multicellular organism process, and inflammatory response (Fig EV1E) and in cellular localizations that include extracellular space, intrinsic component of membrane, and secretory vesicles and granules (Fig EV1F).

We next analyzed the covariation between mRNA levels and PTR ratios of genes across tissues. Among 3,753 genes with valid PTR ratio values in at least 15 tissues and with a strong variation of mRNA levels and PTR ratios (standard deviations greater than three-fold), 31 genes displayed positive and 569 genes displayed negative correlation (FDR < 0.1) between these two measures. Also, Multi-Omics Factor Analysis (Argelaguet et al, 2018) (Materials and Methods) showed that the latent factors explaining 60% of the across-tissue variance of mRNA levels were only able to explain 35% of the variance in PTR ratios (Fig 1E). Moreover, most of these latent factors were specific to either mRNA or PTR ratio level indicating that joint likelihood optimization failed to find significant factors that capture the shared covariation between mRNA and PTR ratio across tissues (Fig 1F). Together, these observations suggest that a substantial amount of the regulation of PTR ratios is independent of mRNA level regulation.

## Tissue specificity of RNA binding proteins

We next investigated tissue-specific expression of RNA binding proteins, which are among the major factors controlling protein translation. Overall, 1,233 out of 11,575 inspected genes were among the 1,542 RNA binding proteins manually curated by Gerstberger et al (2014). Of these, 825 RBPs were measured in all 29 tissues (Appendix Fig S4A). According to tissue specificity scores defined by Gerstberger et al, 135 out of 1,233 RBPs were defined as being tissue-specific (Table EV5) based on our RNA-Seq dataset, which was consistent with the general observation that the majority of the RBPs are ubiquitously expressed and typically at higher levels than average cellular proteins (Vaquerizas et al, 2009; Gerstberger et al, 2014; Kechavarzi & Janga, 2014). The 135 tissue-specific RBPs

were significantly enriched in spermatogenesis, the multi-organism reproductive process, DNA modification, and meiotic nuclear division and localized in germ plasm, pole plasm, and P granule (FDR < 0.1; Appendix Fig S4B and C).

**Sequence features predictive of protein-to-mRNA ratio**

To identify and quantify sequence determinants of protein-to-mRNA ratio, we derived a model predicting tissue-specific PTR ratios from mRNA and protein sequence alone. The model is a multivariate linear model that includes a comprehensive set of mRNA-encoded and protein-encoded sequence features known to modulate translation initiation, elongation, and termination, as well as protein stability (Fig 2A and B, Materials and Methods, Table EV6). The model also includes the GC content and the length of each gene region in order to capture technical biases. Furthermore, the model includes sequence features that we identified *de novo* through systematic association testing between either median PTR ratios across tissues or tissue-specific PTR ratio fold-changes relative to the median, and the presence of k-mers, i.e., subsequences of a predefined length $k$, in the 5′ UTR, the coding sequence, the 3′ UTR, and the protein sequence (Materials and Methods). We report our findings below, going from 5′ to 3′. The effects of each sequence feature on PTR log-ratio are estimated using the joint model, thereby controlling for the additive contribution of all other sequence features (Table EV7). We underscore that these reported effects are estimated with a multivariate model from observational data. Hence, they may or may not reflect the effects of creating or removing a single sequence feature in a given gene because they are estimated from observational data, because of potential regression artifacts such as spurious correlations and regression-toward-the-mean, and also because of the simplifying modeling assumption that the regulatory elements function independently of each other.

*mRNA 5′ UTR sequence features*
Negative minimum RNA folding energy in 51-nt sliding windows, a computational proxy for RNA secondary structure, associated with a lower PTR ratio around the start codon (Fig 2C, up to 9% decrease, FDR < 0.1, Materials and Methods). This negative association is in agreement with mechanistic studies in *E. coli* showing that secondary structures around the start codon impair translation by sterically interfering with the recruitment of the large ribosome subunit (Kudla *et al*, 2009). In contrast, negative minimum folding

energy in 51-nt windows associated positively with the PTR ratio about 48 nt downstream of the start codon (Fig 2C, up to 7% increase, FDR < 0.1). This positive association is consistent with experiments showing that hairpins located downstream of the start codon facilitate start codon recognition of eukaryotic ribosomes *in vitro* (Kozak, 1990), presumably by providing more time for the large ribosome subunit to be assembled.

Investigating every 3- to 8-mer in the 5′ UTR, while controlling for occurrence of other k-mers, revealed 6 k-mers significantly associated with median PTR ratio across tissues, as well as 19 further k-mers associated with tissue-specific PTR ratio at a false discovery rate (FDR) < 0.1 (Materials and Methods). The 6 k-mers that were significantly associated with median PTR ratio across tissues include AUG, the canonical start codon, for which at least one occurrence out-of-frame relative to the main ORF associated with about 18–33% lower median PTR ratios across tissues (Fig 2D). This observation is consistent with previous reports that out-of-frame AUGs in the 5′ UTR (uAUG; Kozak, 1984) and upstream ORFs (uORF; Morris & Geballe, 2000; Calvo *et al*, 2009; Barbosa *et al*, 2013) associate with lower protein-per-mRNA amounts. No significant associations could be found for the 796 transcripts with only in-frame uAUGs (Fig EV2A). Among 2,483 transcripts with a single uAUG or uORF, a single out-of-frame uAUG is associated with a 20% reduced PTR ratio compared to a single out-of-frame uORF (Fig EV2B), possibly because ribosomes can re-initiate translation downstream with high efficiency after translating a uORF (Morris & Geballe, 2000). These uAUGs are significantly conserved (one-sided Wilcoxon test, $P = 1 \times 10^{-37}$) compared to background flanking regions according to the PhastCons score (Siepel *et al*, 2005) computed across 100 vertebrates (Fig EV2C, Materials and Methods), which is consistent with earlier conservation analyses of AUG triplets in mammalian and yeast 5′ UTRs (Churbanov *et al*, 2005).

While the out-of-frame uAUG associated significantly with decreased PTR ratio in all 29 tissues, the other 24 5′ UTR k-mers showed significant effects on PTR ratio (FDR < 0.1) only in certain tissues (Fig 2E). These 24 k-mers were found in between 215 transcripts (2%) for AGCGGAA and 3,038 transcripts (26%) for GCCGCC (Fig EV3A). To search for possible proteins binding these k-mers, we queried the ATtRACT database (Giudice *et al*, 2016), which is, to our knowledge, the most extensive database of RNA binding motifs and contains 3,256 position weight matrices collected for 160 human RNA binding proteins (Fig EV3A, Materials and

**Figure 2.  Predicting PTR ratios from sequence and 5′ UTR results.**

A   Sequence features of 5′ UTR, coding sequence, 3′ UTR, and protein sequence considered in the model.

B   The predictive model is a multivariate linear model that predicts tissue-specific PTR log-ratios using tissue-specific coefficients for the sequence features listed in (A).

C   Effect of $\log_2$ negative minimum folding energy of 51-nt window on median $\log_{10}$ PTR ratio across tissues corrected for all other sequence features listed in (A) (y-axis, Materials and Methods) versus position of the window center relative to the first nucleotide of the canonical start codon (x-axis) for genes with a 5′ UTR and a coding sequence longer than 100 nt. Statistically significant effects at $P < 0.05$ according to Student's *t*-test and corrected by the Benjamini–Hochberg methods are marked in red.

D   Effect estimate (dot) and 95% confidence interval (bar) of the presence of at least one out-of-frame AUG in 5′ UTR on $\log_{10}$ PTR ratio corrected for all other sequence features listed in (A) (y-axis, Materials and Methods) per tissue (x-axis).

E   Estimated effect of PTR ratio in each tissue (row) of the 25 5′ UTR k-mers (column) associating with either median PTR ratio across tissues or tissue-specific gene-centered PTR ratios. Color scale ranges from blue (negative effect) to red (positive effect). Gray marks non-significant (*FDR* ≥ 0.1) associations.

F   Average 100-vertebrate PhastCons score (y-axis, Materials and Methods) per position relative to the exact motif match instances in 5′ UTR (x-axis) for three example k-mers that are significantly predictive of PTR ratios in specific tissues. *P*-values assess significance of the average 100-vertebrate PhastCons scores at the motif sites compared to the two 10-nucleotide flanking regions (Materials and Methods). The motif logos are constructed using all matches of the considered k-mer up to one mismatch in the 5′ UTR sequences.

   

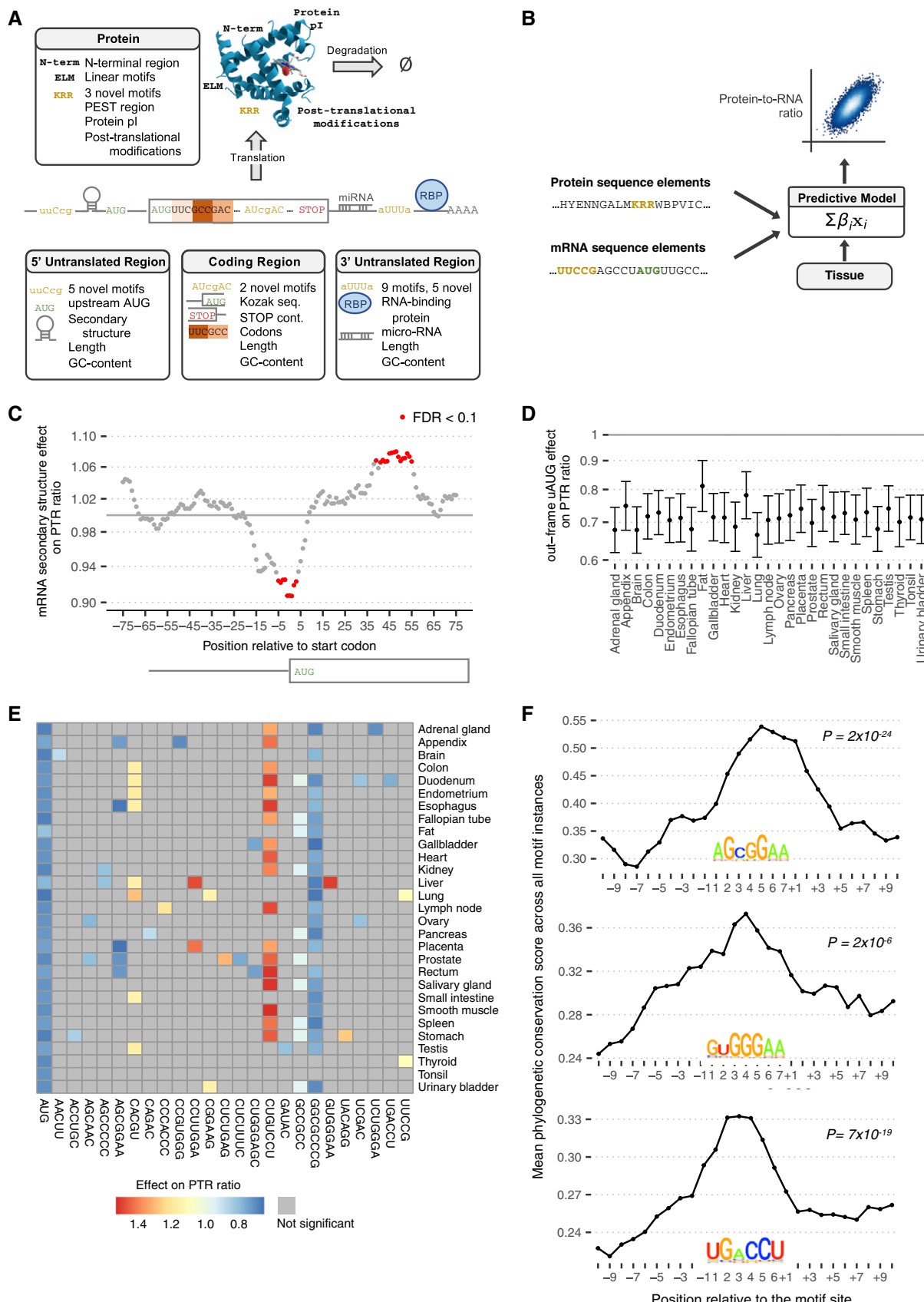

**Figure 2.**

Methods). However, no obvious association between these k-mers and RNA binding motifs could be drawn as most of the matches remain very distant (ATtRACT quality score < 0.1). A potential reason is that the ATtRACT database covers a small fraction of all human RBPs, which could consist of more than 1,500 proteins (Gerstberger *et al*, 2014). Nonetheless, 11 out of 24 of our k-mers were significantly more conserved than their flanking regions (FDR < 0.1, Fig 2F, Appendix Fig S5) and 12 showed significant enrichment for Gene Ontology (GO) terms (Appendix Fig S6), supportive for a potential regulatory role. The appendix provides a comprehensive description of these results.

### Start and stop codon context

Significant associations of individual nucleotides with the PTR ratio were detected in the [−5,6] nt interval around the start codon and in the [−4,6] nt interval around the stop codon in at least one tissue (FDR < 0.1; Fig EV2D and E). At nearly every position of the start codon context, the nucleotide of the consensus sequence gccRccAUGG (Kozak, 1986) showed the strongest association, indicating selection for efficient start codon recognition. The strongest effects were found at the third position upstream of the start codon (27% lower PTR ratio for C than for the consensus A), recapitulating mutagenesis data (Kozak, 1986), and at the second nucleotide downstream of the start codon (23% lower PTR ratio for A than for the consensus C). Moreover, effects of the start codon context on the PTR ratio were largely independent of the tissue (Fig EV2D) consistent with a ubiquitous role of the start codon context likely due to structural interaction with the ribosome (Svidritskiy *et al*, 2014).

The opal stop codon UGA was significantly associated with the lowest median PTR ratio having in median 15% lower PTR ratios than the ocher stop codon UAA (Fig EV2F; *P* = 1.2 × 10$^{-5}$). Around the stop codon, the two most influential positions were the +1 nucleotide at which a C associated with 15% lower PTR ratios than the consensus G, and the -2 nucleotide, at which a G associated with 19% lower PTR ratios than the consensus A in median across tissues (Fig EV2E). The inhibitory effect of a C at the +1 nucleotide, which was observed for all three stop codons (Fig EV2G), is in line with previous studies in prokaryotes and eukaryotes (Bonetti *et al*, 1995; McCaughan *et al*, 1995; Poole *et al*, 1995; Tate *et al*, 1996).

Also, structural data show that a C following the stop codon interferes with stop codon recognition (Brown *et al*, 2015), thereby leading to stop codon read-through. Moreover, our data indicate that the nucleotide at the −2 position, which is also reported to be highly biased in *E. coli* (Arkov *et al*, 1993), is significantly associated with PTR ratio and deviation from the consensus nucleotide A is associated with a reduced PTR ratio. Altogether, the start and stop codon contexts demonstrate the sensitivity of the PTR ratio analysis in detecting contributions to translation down to single-nucleotide resolution.

### Amino acid and synonymous codon usage

Codon frequency can affect PTR ratios in several ways. On the one hand, synonymous codon usage modulates translation efficiency (Gardin *et al*, 2014; Yu *et al*, 2015; Weinberg *et al*, 2016; Yan *et al*, 2016; Hanson & Coller, 2018). On the other hand, amino acid identity affects translation (Wilson *et al*, 2016; Hanson & Coller, 2018) and protein half-life (Fang *et al*, 2014; Zecha *et al*, 2018). Among all investigated sequence features, amino acid frequency had the largest predictive power for PTR ratio in every tissue (explained variance between 12 and 17%, median 15%; Figs 3A and EV3B). We defined the amino acid effect on PTR ratio as the PTR ratio fold-change associated with doubling the frequency of an amino acid in a gene (Materials and Methods). The amino acid effects were large with a twofold increase in amino acid frequencies associating with 40% lower PTR ratio for serine (S) and 50% higher PTR ratio for aspartic acid (D) (Fig 3A, Materials and Methods). Codon frequency, which inherently encodes amino acid frequency and synonymous codon usage, increased that explained variance on average by only 1% (explained variance between 13 and 20%, median 16%; Figs 3B and EV3B). We defined the protein-to-mRNA ratio adaptation index (PTR-AI) as the PTR ratio fold-change associated with doubling the frequency of a codon in a gene (Materials and Methods). Synonymous codons coding for the same amino acids displayed different PTR-AIs (Fig 3B). Moreover, the PTR-AI of individual codons showed consistent amplitudes and directions across tissues (Fig 3B), which contests the hypothesis of widespread tissue-specific post-transcriptional regulation due to a varying tRNA pool among different tissues (Plotkin *et al*, 2004; Dittmar *et al*, 2006). We observed differences of codon frequency in the 5′ end of the coding sequence

**Figure 3.   mRNA coding region sequence features results.**

A   Distribution of the amino effect on PTR ratio per tissue, which is the PTR ratio fold-change associated with doubling the frequency of the amino acid (Materials and Methods). Shown are the quartiles (boxes and horizontal lines) and furthest data points still within 1.5 times the interquartile range of the lower and upper quartiles (whiskers).

B   Same as (A) for codons (PTR-AI). The codons are grouped by the amino acid they encode and are sorted first by increasing amino acid effect, then by increasing synonymous codon effect.

C   Median codon decoding time (transformed to z-scores) across 17 independent ribosome profiling datasets (*y*-axis), grouped per amino acid (*x*-axis). Red dots display the average amino acid decoding time (Materials and Methods). Amino acid types explain 70% of the variation in the decoding times of 61 codons.

D   Median codon decoding time estimates (z-scores) across 17 independent human Ribo-Seq datasets (*x*-axis) significantly negatively correlate with average PTR-AI across tissues (*y*-axis).

E   Same as (A) for the distribution of the amino acid effect on protein half-lives, which is the protein half-life fold-change associated with doubling the frequency of the amino acid, for five different cell types: HeLa cells (Zecha *et al*, 2018), B cells, NK cells, hepatocytes, and monocytes (Mathieson *et al*, 2018).

F   Amino acid effect on protein half-lives (*x*-axis) significantly positively correlates with amino acid effect on PTR ratio (*y*-axis).

G   Correlation network of the amino acid or codon frequency when applicable on PTR ratio, codon decoding time, and protein half-life. Significant Spearman correlations (*P* < 0.05) are found between the effects on PTR ratio and codon decoding time, and between the effects on PTR ratio and protein half-life but not between codon decoding time and protein half-life.

H   Codon tRNA adaptiveness (*x*-axis), a widely used codon optimality metric, does not significantly correlate with PTR-AI (*y*-axis), which may be a new optimality metric reflecting the combined effect of amino acid and synonymous codon usage on protein synthesis and degradation.

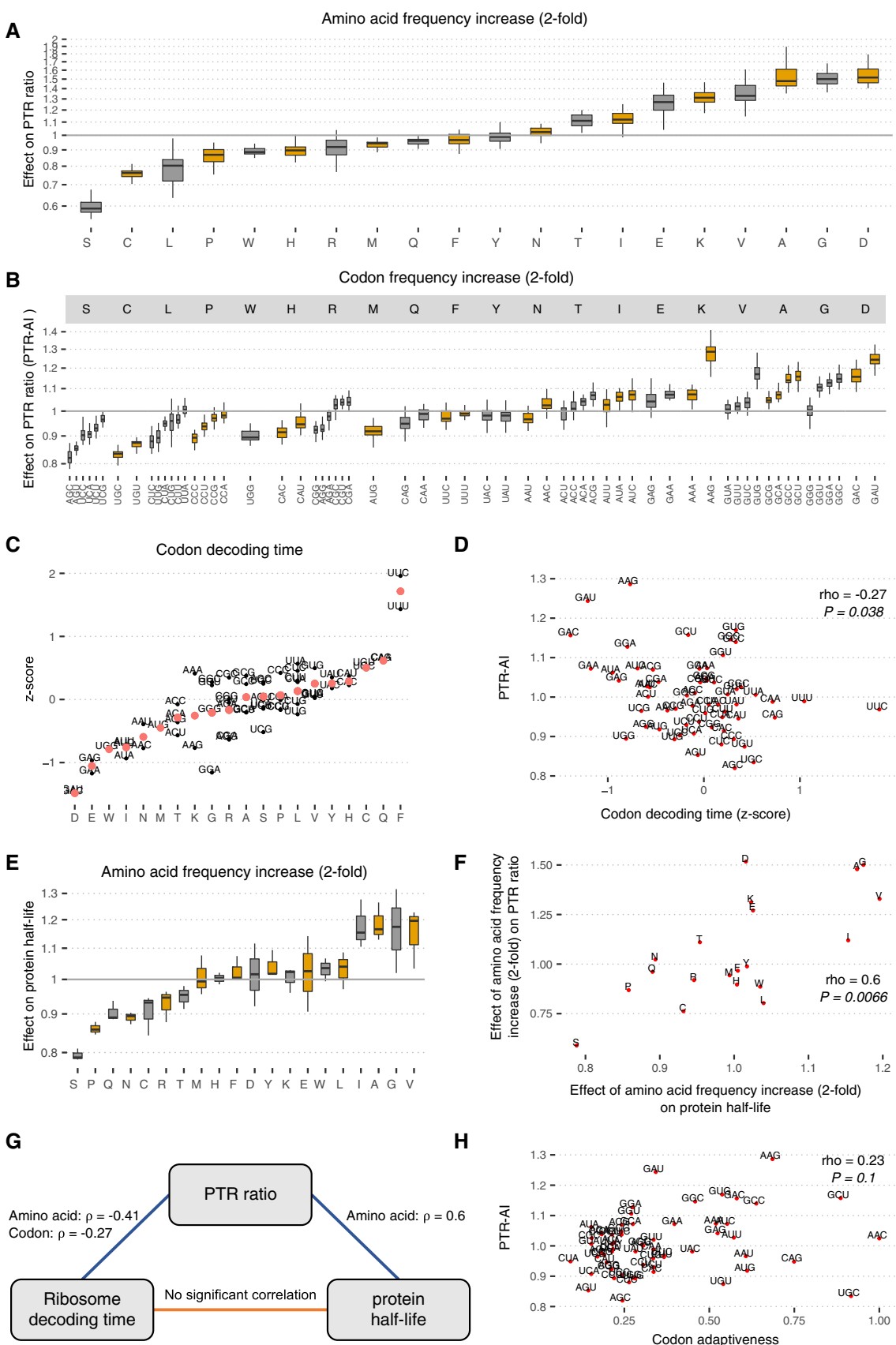

Figure 3.

compared to the rest of the coding region (Fig EV3C). However, PTR-AI correlated significantly between these two regions (Fig EV3D). We therefore did not distinguish the 5′ end region of the coding sequence from the rest of the coding sequence when considering codon frequencies in our model.

To relate the amino acid and synonymous codon effects to translation and protein degradation, both of which contribute to PTR ratios, we first investigated codon decoding times, whereby long decoding times would lead to lower translation output (Gustafsson *et al*, 2012; Gardin *et al*, 2014; Ingolia, 2014; Yu *et al*, 2015; Weinberg *et al*, 2016; Yan *et al*, 2016; Hanson & Coller, 2018). We considered codon decoding time as the typical time ribosome takes to decode a codon (Dana & Tuller, 2014), also sometimes referred to as ribosome dwell time (O'Connor *et al*, 2016). We computed (Materials and Methods) median codon decoding times across 17 ribosome profiling datasets (Dana & Tuller, 2015; O'Connor *et al*, 2016). Notably, amino acid identity explained 70% median codon decoding time variance (Fig 3C), consistent with the dominant role of amino acids on PTR ratio. The strong association between amino acid identity and codon decoding time may be in part reflecting that the amino acid content of the nascent polypeptide chain influences translation elongation (Charneski & Hurst, 2013). Moreover, PTR-AIs correlated significantly negatively with median codon decoding times (Fig 3D, Spearman's $\rho = -0.27$, $P = 0.03$, Fig EV4A). We also found that median PTR-AIs correlated significantly positively with predicted effects of codons on mRNA stability (Materials and Methods) in K562 (Spearman's $\rho = 0.47$, $P = 4.7 \times 10^{-5}$; Fig EV4AB; Schwalb *et al*, 2016), in HEK293 (Spearman's $\rho = 0.48$, $P = 9 \times 10^{-5}$; Fig EV4AB; Schueler *et al*, 2014), and in HeLa Tet-off cells (Spearman's $\rho = 0.52$, $P = 3 \times 10^{-5}$; Fig EV4AB; Tani *et al*, 2012). This agreement of PTR-AIs and predicted effects of codons on mRNA stability is consistent with the fact that codon composition is causally affecting mRNA degradation (Hoekema *et al*, 1987; Presnyak *et al*, 2015; Bazzini *et al*, 2016; Mishima & Tomari, 2016) in a way that is mediated by translation (Radhakrishnan & Green, 2016). Together, these results indicate that PTR-AIs capture the effect of codons on translation.

We then asked whether our amino acid effects on PTR ratios captured the effects of amino acids on protein degradation. To this end, we first performed a linear regression of protein half-lives measured in HeLa cells (Zecha *et al*, 2018), B cells, NK cells, hepatocytes, and monocytes (Mathieson *et al*, 2018) on amino acid frequency (Materials and Methods). We defined the amino acid effect on protein half-life as the protein half-life fold-change associated with doubling the frequency of an amino acid in a gene. The amino acid effects on protein half-life agreed well among these datasets (Fig 3E) with proportions of explained variance varying from 9% for monocytes to 19% for NK cells. Moreover, the amino acid effects on protein half-life significantly correlated with the effects of single amino acid substitutions on protein thermodynamic stability (Dehouck *et al*, 2009, Fig EV5A; Spearman's $\rho = 0.18$, $P = 0.002$) and with amino acid hydrophobicity values (Fig EV5B; Spearman's $\rho = 0.42$, $P = 0.04$), a major force stabilizing the folding of proteins (Nick Pace *et al*, 2014). This suggests that the associations of amino acids with protein half-lives are in part functional and due to the role of amino acids on protein thermodynamic stability, a strong determinant of protein cytoplasmic degradation (Díaz-Villanueva *et al*, 2015).

Overall, the amino acid effects on PTR ratio correlated significantly with both the amino acid effects on protein half-life (Spearman's $\rho = 0.6$, $P = 0.006$; Fig 3F and G) and the average amino acid decoding time (Spearman's $\rho = -0.41$, $P = 0.03$; Fig 3G, Materials and Methods). However, average amino acid decoding times did not correlate significantly with the amino acid effects on protein half-life (Spearman's $\rho = -0.22$, $P = 0.35$). Analogous results were obtained by taking a codon-centric rather than an amino acid-centric point of view. Specifically, PTR-AI correlated significantly with codon effects on protein half-life (Spearman's $\rho = 0.56$, $P = 4.7\text{e-}06$; Materials and Methods) on the one hand, and with codon decoding time (Spearman's $\rho = -0.27$, $P = 0.04$; Fig 3D) on the other hand. However, codon decoding time did not correlate significantly with codon effects on protein half-life (Spearman's $\rho = -0.09$, $P = 0.45$). Hence, PTR-AI appears to capture a combination of apparently independent effects of codon frequency on translation elongation and amino acid frequency on protein stability. Notably, PTR-AI did not correlate well with previous codon optimality measures, including the frequency of codons in human coding sequences (Appendix Fig S7, Spearman's $\rho = 0.2$, $P = 0.11$, Materials and Methods) and species-specific codon absolute adaptiveness (Sabi & Tuller, 2017; Fig 3H, Spearman's $\rho = 0.23$, $P = 0.1$), which are based on genomic or transcriptomic data and strong modeling assumptions. Altogether, these results indicate that a PTR ratio-based measure of codon optimality, which captures the combined effects of protein production and degradation, is an attractive alternative to existing codon optimality measures and could help resolving some of the debates about the role of codon optimality in human cells.

### Protein sequence features

Our model includes further protein sequence features beyond the mere amino acid composition. Although the N-terminal amino acid (which is known to affect protein stability via the N-end rule pathway) significantly associated with the PTR ratio (Appendix Fig S8), the N-terminal amino acid was not significant in the joint model, possibly because the effect was confounded with the start codon context. A recent study by Kats and colleagues (Kats *et al*, 2018) in yeast indicated that the mean hydrophobicity of the first 15 amino acids plays a more important role in protein stability than the N-end rule pathway. We observed that mean hydrophobicity of the first 15 amino acids significantly associated with the PTR ratios of 8 tissues (3% higher PTR ratio on average, FDR < 0.1; Fig 4), however positively, in apparent contradiction with its negative effect on protein stability in yeast (Kats *et al*, 2018). This may be due to the multiple roles of the 5′ end of the coding region in gene expression regulation, which also includes a role in translation (Tuller & Zur, 2015). We also considered protein surface charge–charge interactions because they can affect protein stability (Samantha *et al*, 2006; Chan *et al*, 2012), and because the charged polypeptides in the ribosome exit tunnel can influence ribosome elongation speed (Requião *et al*, 2017). Consistently, we observed that a one unit increase in the protein isoelectric point had a significant negative association with the PTR ratio (median 5%) in several tissues (Materials and Methods, Fig 4; FDR < 0.1). Our analysis also confirmed, genome-wide, the negative effect on PTR ratios of PEST regions, which are degrons that are rich in proline (P), glutamic acid (E), serine (S), and

threonine (T) (Rogers *et al*, 1986) that were present in 4,592 proteins (Materials and Methods), and estimated its median effect across tissues to a 26% lower PTR ratio (Fig 4; FDR < 0.1).

*De novo* motif searching revealed two 2-mers and one 3-mer associating with lower PTR ratios (11, 14, and 7% median effects for CG, KRR, and NS, respectively, Fig 4, FDR < 0.1). The effect for KRR is consistent with the association of stretches of positively charged amino acids directly upstream of high ribosome occupancy peaks in ribosome footprint data, suggesting that positively charged amino acids slow down translation (Charneski & Hurst, 2013). However, lysine (K) and arginine (R) are also the two amino acids recognized by cleavage sites of trypsin, the enzyme used to digest proteins prior to mass spectrometry. Although K and R as single amino acids do not stand out as negatively associated with the PTR ratio (Fig 3A), we cannot exclude a technical bias for the negative association of the 3-mer KRR with PTR ratios. Furthermore, we identified 6 linear protein motifs out of the 267 motifs from the ELM database (Dinkel *et al*, 2016) using a feature selection method (Materials and Methods). These 6 linear protein motifs contained 4 nuclear localization signals of the ELM database which associated negatively with PTR ratios. It is unclear why these four nuclear localization signals were associated negatively with PTR ratio even though there is no significant PTR ratio difference between

nuclear (GO:0005634) and non-nuclear proteins (Appendix Fig S9). One possibility is that these linear motifs are destabilizing elements. Indeed, these 6 linear protein motifs were significantly associated with shorter protein half-lives (Appendix Fig S10). Also, nuclear proteins with the four nuclear localization signals were associated with shorter half-lives compared to nuclear proteins without these signals (Appendix Fig S10). We also note that these linear motifs are KR-rich. Similar to the association of the 3-mer KRR, this could reflect either that stretches of positively charged amino acid slow down translation or a technical bias due to the usage of trypsin as the protein digestion enzyme (Materials and Methods, Fig 4).

### mRNA 3′ UTR sequence features

*De novo* motif searching in the 3′ UTR revealed 20 k-mers significantly associated with median PTR ratios across tissues or with tissue-specific PTR ratio (FDR < 0.1; Fig 5A and B). This recovered 4 well-known mRNA motifs: the polyadenylation signal AAUAAA (Proudfoot, 1991), the AU-rich elements UAUUUAU (Kruys *et al*, 1989; Qi *et al*, 2012) and AUUUUUA (Ma *et al*, 1996), and the binding site of the Pumilio family of proteins UGUAAAUA (Parisi & Lin, 2000). The polyadenylation signal AAUAAA associated with between 13 and 28% increased PTR ratio across tissues (median

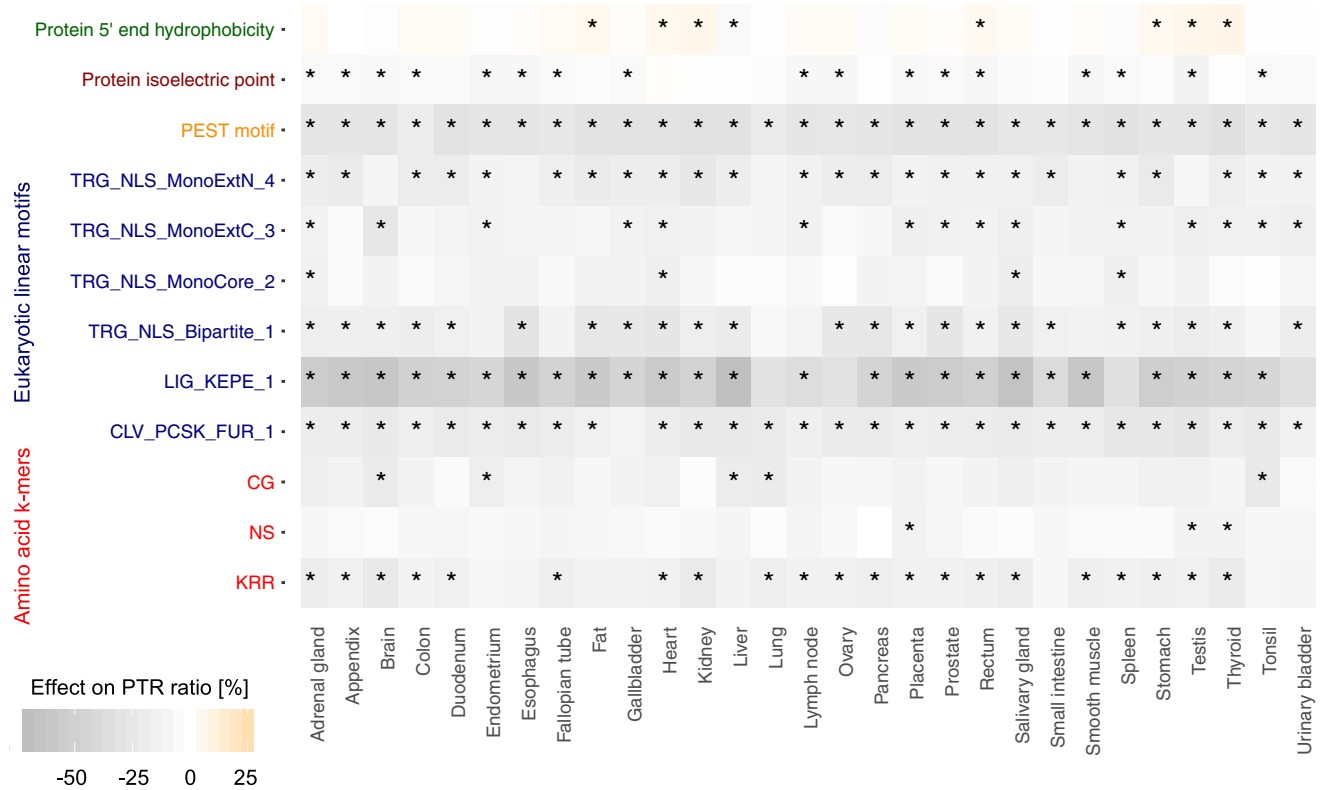

**Figure 4. Protein sequence features.**

Heatmap showing tissue-specific associations of protein sequence features with higher [0, + 25] (red gradient) or lower [−70, 0] (blue gradient) PTR ratios. Stars represent tissue-specific significance of the sequence feature with FDR < 0.1. Eukaryotic protein motif acronyms are CLV_PKCS_FUR_1 (Furin (PACE) cleavage site), LIG_KEPE_1 (Sumoylation site), TRG_NLS_BIPARTITE_1 (classical bipartite nuclear localization signal), and three classical monopartite nuclear localization signals: TRG_NLS_MonoCore_2, TRG_NLS_MonoExtC_3, and TRG_NLS_MonoExtN_4.

21%, FDR < 0.1; Fig 5A), consistent with one role of polyadenylation signals in translation (Piqué *et al*, 2008). The AU-rich element UAUUUAU was found in 3,158 genes (27%) and associated with lower PTR ratios by about 9% consistently across tissues, in agreement with its function in mRNA destabilization and translational silencing (Kruys *et al*, 1989; Qi *et al*, 2012). The Pumilio motif UGUAAAUA was found in 1,320 genes (11%) and is the binding target of members of the Pumilio family of proteins which regulate translation and mRNA stability in a wide variety of eukaryotic organisms (Parisi & Lin, 2000). In addition to these four evolutionarily conserved motifs (Appendix Fig S11), we identified 7 motifs, namely ACCAAA, CCAAAG, CUCAGG, GGGCUGCG, GGAGCC, GGCCCUG, and UUCUGAG; these are also significantly conserved with respect to the background flanking regions (Appendix Fig S10). While some of these conserved motifs were not previously reported in the literature, some of the obtained k-mers may possibly be the binding motifs of RBPs with a post-transcriptional role. One notable example is the k-mer ACACUA, which matches a recognition site of the QKI protein according to the ATtRACT database (quality score = 1.0), which is highly enriched in the brain (Human Protein Atlas; Uhlen *et al*, 2015) and important for myelinization (Aberg *et al*, 2006), mRNA stability, and protein translation (Teplova *et al*, 2013). Another example is the well-conserved ACCAAA, present in 3,655 genes (32%), possibly being the target motif of RBMX (ATtRACT quality score = 1.0) which plays several roles in the regulation of post-transcriptional processes (Kanhoush *et al*, 2010). The appendix provides a full analysis per motif based on their number of occurrences, phylogenetic conservation scores (Appendix Fig S11), and a gene set enrichment analysis for the genes having the consensus motif (Appendix Fig S12).

### An interpretable model explaining PTR ratios from sequence

The multivariate linear model combining all these sequence features predicted PTR ratios at a median relative error of 3.2-fold on held-out data (10-fold cross-validation), which is small compared to the overall variation of PTR ratios (200-fold for the 80% equi-tailed interval). This model explained 22% (median across tissues) of the variance (Fig 5C). Moreover, we observed that the predicted PTR ratios moderately positively correlated with the mRNA levels (Fig 5D; Spearman's ρ = 0.26). Hence, our model supports the hypothesis that highly transcribed genes also have optimized sequences for post-transcriptional up-regulation, hence yielding higher amounts of proteins, which is consistent with earlier work by Vogel and colleagues (Vogel *et al*, 2010). Combining these sequence features together with the mRNA profiles in a single linear model explained 58% of the variance of tissue-specific protein levels in average (minimum 49% in pancreas, maximum 63% in liver; Materials and Methods), increasing the proportions of variance of tissue-specific protein levels explained with mRNA profiles alone (shown in Fig 1B) by 10% in average ($P = 3 \times 10^{-9}$, Wilcoxon test).

### Extended model with experimentally characterized elements

There are thousands of further sequence elements that could play a role in controlling the PTR ratios, including the binding sites of any of the 2,599 catalogued human miRNAs (Chou *et al*, 2018), the binding sites of the estimated 1,542 RNA binding proteins (Gerstberger *et al*, 2014), and elements subject to mRNA modifications and post-translational modifications of certain amino acids.

In this context, derivation of a more comprehensive yet interpretable model of PTR ratio from sequence is difficult. One reason is that the sequence determinants driving the binding of these factors and these modifications are poorly charted. Another reason is that binding sites of RBPs and miRNAs often co-occur due to cooperative and competitive binding (Jacobsen *et al*, 2010; Chang & Hla, 2011; Jiang & Coller, 2012; Ciafrè & Galardi, 2013), which makes untangling the effects of individual sequence elements difficult. Nevertheless, in order to explore the degree to which the prediction of the PTR ratio from sequence could be improved in principle, we considered a model that was not based on sequence alone, rather also including experimental characterization of such interactions and modifications of mRNA and proteins. This extended model included (i) *N6*-methyladenosine (m6A) mRNA modification, an abundant modification enhancing translation (Wang *et al*, 2015); (ii) binding evidence for 296 miRNAs from the miRTarBase database (Chou *et al*, 2018) with more than 200 targets in our dataset (Materials and Methods); (iii) whether proteins are part of protein complexes, which is known to stabilize proteins (Mueller *et al*, 2015; Ishikawa *et al*, 2017); (iv) binding evidence to 112 RNA binding proteins (RBPs) (Van Nostrand *et al*, 2016); and (v) phosphorylation, methylation, acetylation, SUMOylation, and ubiquitination of certain amino acids (Hornbeck *et al*, 2015). This analysis showed that with the inclusion of these experimentally characterized features, the proportion of variance of PTR ratio increased to a median across tissues of 27% (Fig 5E; min 24%, max 31%). Moreover, combining the extended set of features together with the mRNA profiles in a single linear model explained 62% of the variance of tissue-specific protein levels in average (minimum 53% in pancreas, maximum 68% in tonsil; Materials and Methods). However, these increased proportions of variance explained do not imply that these experimentally characterized features are not driven by regulatory elements encoded in sequence. Rather, they may reflect that our primary regression of PTR ratio on sequence features was not powerful enough to capture those underlying, potentially complex, regulatory sequence elements.

Analysis of explained variance of individual feature groups indicated that amino acid frequency alone explained on average 15% of the variance in PTR ratios (min 12%, max 15%; Figs 5E and EV5C). This is followed by protein acetylation sites, binding sites of 112 RBPs (Van Nostrand *et al*, 2016), CDS length, protein ubiquitination sites, and linear protein motifs (Figs 5E and EV5D). These results suggest that sequence elements affecting protein stability may be the dominant features predictive of PTR ratios. In line with this possibility, we observed that the explained variance in PTR ratio by these sequence features highly correlated (Spearman's ρ = 0.59, $P = 0.001$) with their explained variances in protein half-lives (Fig 5F) in five cell types (Mathieson *et al*, 2018; Zecha *et al*, 2018).

The proportion of variance in PTR ratio explained by the binding evidence to 112 RNA binding proteins (Van Nostrand *et al*, 2016) varied from 3 to 6% across tissues (median 5%), while 150 latent variables of 296 miRNAs' binding evidence explained on average only 1%. Overall, these RBPs appeared to be ubiquitously expressed since 81 out of the 112 RBPs (77%) were detected expressed at the proteome level and at the mRNA level in all tissues. Ubiquitous expression of RBPs and the frequent co-binding of RBPs and miRNAs may be two reasons why tissue-specific effects of RBP binding on PTR ratio did not show significant correlations with the

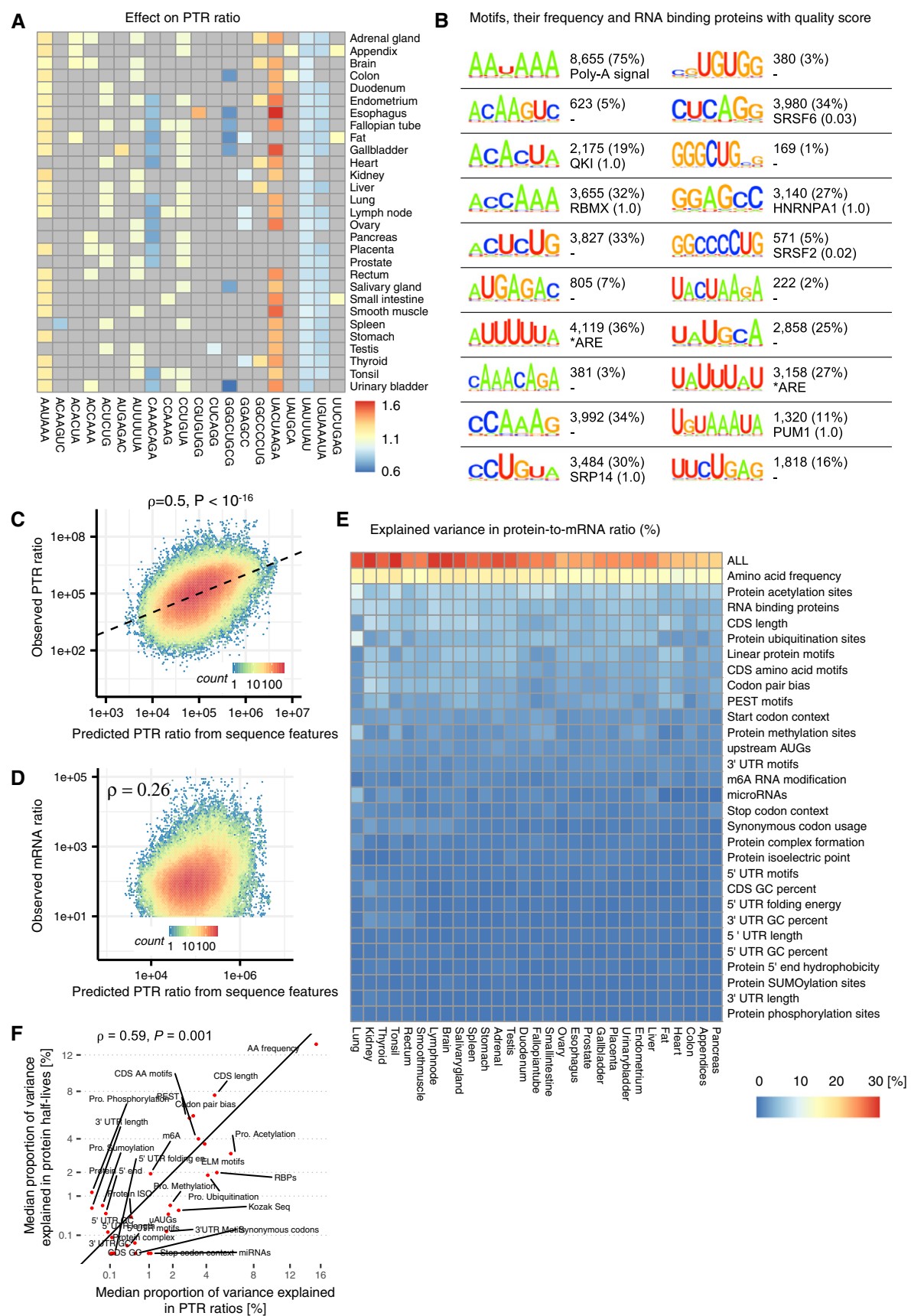

Figure 5.

**Figure 5.   3′ UTR results and model summary.**

A   Estimated effect of PTR ratio in each tissue (row) of the 20 3′ UTR k-mers (column) associating with either median PTR ratio across tissues or tissue-specific gene-centered PTR ratios. Color scale ranges from blue (negative effect) to red (positive effect). Gray marks non-significant ($FDR \geq 0.1$) associations.

B   First and third columns: motif information content logos for the 20 k-mers of panel A, obtained by motif consensus sequence search in 11,575 5′ UTR sequences allowing for one mismatch (Materials and Methods). Second and fourth columns: number and percentage of transcripts consensus motif sequence among the 11,575 transcripts (first line) and best significantly matching RNA binding protein motif of the database ATtRACT (Giudice *et al*, 2016) together with the ATtRACT motif quality score Q (value between 0 and 1, the higher the better; Materials and Methods).

C   Observed PTR ratios of all tissues (*y*-axis) versus predicted PTR ratios by the interpretable sequence model (*x*-axis) which includes 18 sequence feature groups representing 204 post-transcriptional regulatory elements.

D   Observed mRNA levels of all tissues (*y*-axis) correlates with predicted PTR ratios by the interpretable sequence model (*x*-axis) which includes 18 sequence feature groups representing 204 post-transcriptional regulatory elements. This observation supports the hypothesis that genes that are highly transcribed are also optimized for post-transcriptional regulation leading to higher protein levels.

E   Proportion of variance in tissue-specific PTR ratios explained ($R^2$) by separate linear models representing one sequence feature group in each tissue. The first row (labeled "ALL") corresponds to the linear model combining all of the features displayed in the consecutive rows.

F   Median proportion of variance in tissue-specific PTR ratios explained (*x*-axis, $R^2$) by each sequence feature group shown in (D) highly correlates with median proportion of variance explained in protein half-lives of five different cell types (*y*-axis). Most of the explained variance in PTR ratios is dominated by sequence elements that are highly predictive of protein half-lives. The proportion of variance explained by each sequence feature group is shown in Fig EV5.

corresponding tissue-specific RBP expression levels (Fig EV5C, Appendix Fig S13). Nevertheless, in 16 of these RBPs, there was a significant difference between their across-tissue covariation with their target and non-target genes (Appendix Fig S14, Materials and Methods). The binding of these regulatory elements was among the top mRNA features explaining the tissue-specific mRNA levels and mRNA half-lives of three different cell types (Fig EV5E and F). The binding of the considered 112 RBPs explained on average 18% (min 13%, max 21%) and features representing miRNA binding explained on average 5% (min 4%, max 7%) of the variance in tissue-specific mRNA levels (Fig EV5E). Consistent with that, RBP binding explained on average 17% of the variance (min 12%, max 22%) in mRNA half-lives of K562, HEK293, and HeLa Tet-off cells (Fig EV5F). Likewise, features representing miRNA binding explained 5% (median, min 4%, max 5%) of the variance in mRNA half-lives of these three cell lines. Altogether, the differences and similarities in the explained variances of mRNA levels, mRNA half-life, PTR ratio, and protein half-life suggest that the RBPs and miRNAs considered in our model may be more effective in regulating mRNA stability rather than PTR ratios.

Of note, the proportion of variance explained is driven by the combination of effect size, frequency, and variability of the features across genes. Hence, sequence features which play a crucial role for translation, like the Kozak sequence, can only explain 3% of the genome-wide PTR ratio variation by itself because it is already optimized for most of the genes in the genome. Also, the 5′ and 3′ UTR motifs explain a small fraction of the variance between genes although their effect size can be large (Figs 2E and F and 5A and B) because they typically occur in a small number of genes.

### Independent confirmation of the model

As a starting point to assess the validity of our model and the derived predictions, we employed a threefold approach, consisting of the confirmation of the prediction with an independent transcriptomic and proteomic dataset, a reporter assay measuring the dependence of the expression of a reporter protein on the presence of the 5′ or 3′ UTR sequence motifs, and an immobilized mRNA affinity competition-binding assay to identify motif-specific RNA binding proteins and to measure their interaction strength. The effect of individual sequence features estimated on an independently generated

dataset comprising matched RNA-Seq and proteomics data on 2,854 genes from six patient-derived fibroblasts (Kremer *et al*, 2017) agreed well with the median effects estimated across the 29 tissues (Materials and Methods, Spearman's $\rho = 0.57$, $P < 2.2 \times 10^{-16}$). This was also true when restricting the analysis to the codons (Spearman's $\rho = 0.7$, $P < 2.2 \times 10^{-16}$; Fig 6A), indicating that PTR-AI is reproducible across datasets.

Next, we assessed the effects of motifs in a dual reporter assay in which the nine tested motifs (Tables EV8 and EV9) were inserted in the 5′ UTRs or 3′ UTRs of Gaussia luciferase constructs (Materials and Methods). The same plasmid also expressed a secreted alkaline phosphatase as control. This assay showed significant effects for two positive controls: the out-of-frame upstream AUG and the out-of-frame upstream ORF, i.e., an upstream AUG with an in-frame stop codon within the 5′ UTR (Fig 6B; $P < 0.0001$). For the remaining tested motifs, control constructs containing scrambled versions of the tested motif were also assayed (Appendix Figs S15 and S16, Table EV10). Two tested motifs (UUCCG and CUGUCCU) showed significant effects in the direction predicted by the model (Fig 6C and D, FDR < 0.1, Materials and Methods). Most motifs had small predicted effects, so that significance was difficult to attain in such assays. Taking this into account, four further motifs, including two positive controls, the AU-rich 3′ UTR motif UAUUUAU and the Pumilio response elements, as well as the new motifs CCCACCC and GGCCCCUG, showed effects consistent with the model prediction (Appendix Figs S15 and S16, Materials and Methods) both in direction and in amplitude.

Lastly, we investigated whether one of these motifs, the 5′ UTR motif CUGUCCU, was a potential recognition site of RNA binding proteins. To this end, we performed a series of competition-binding assays with motif-containing RNA oligomers immobilized on Sepharose beads to capture RNA binding proteins from HEK293 cell extracts in the presence of different concentrations of free motif-containing ligands. The captured RNA binding proteins were analyzed by label-free quantitative proteomics. The resulting data allowed the estimation of EC50 values and dissociation constants ($K_d$) akin to affinity chromatography-based chemical proteomics approaches (Bantscheff *et al*, 2007; Médard *et al*, 2015). The assay was also performed for two positive controls, the polyadenylation signal AAUAAA and the AU-rich element UAUUUAU. Reversed or randomized sequences were used as negative controls. Example

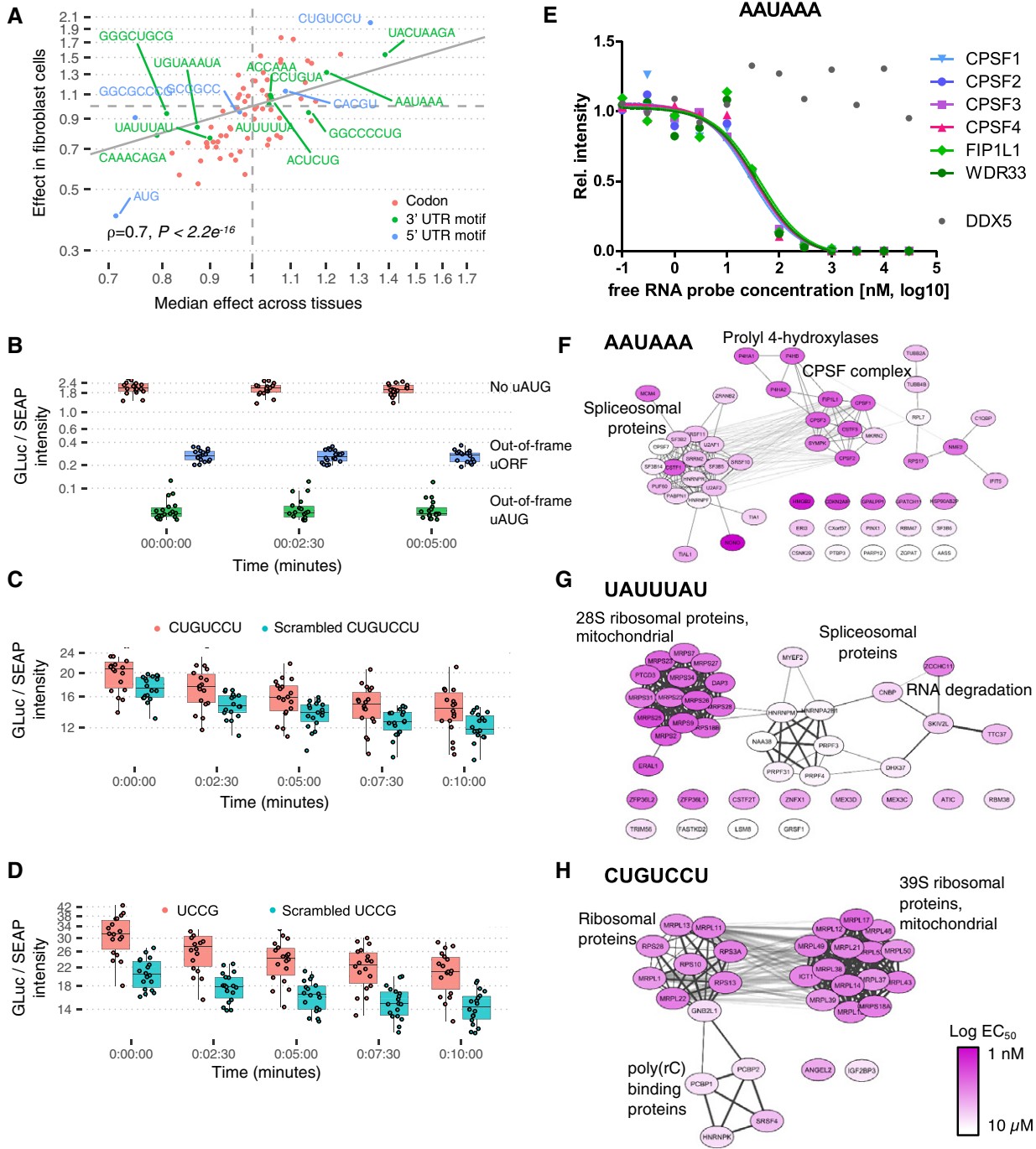

**Figure 6. Independent validations.**

A    Comparison of codon (red), 3′ UTR (green) and 5′ UTR (blue) motif median effects on PTR ratio across 29 human tissues (x-axis) and effects on PTR ratios in an independent matched proteome and transcriptome dataset (Kremer *et al*, 2017). Because one does not expect effect of tissue-specific motifs to necessarily reproduce in Fibroblasts, the plot is restricted to the motifs that show significant association with PTR ratio in at least five tissues.

B    Reporter assay of the AUG in 5′ UTR. Ratio of GLuc over SEAP intensities normalized per experiment (y-axis, n = 18, Materials and Methods) per time point (x-axis) and construct: no insertion (pink), inserted out-of-frame AUG (green), and inserted uORF, i.e., inserted AUG with an inserted stop codon in-frame in the 5′ UTR (blue). Shown are the quartiles (boxes and horizontal lines) and furthest data points still within 1.5 times the interquartile range of the lower and upper quartiles (whiskers). Original GLuc over SEAP intensities for all tested motifs in Appendix Figs S15 and S16.

C    As in (B) for inserted 5′ UTR motif CUGUCCU (pink) or a scrambled version of it (blue, UUUGCCC).

D    As in (B) for inserted 5′ UTR motif UUCCG (pink) or a scrambled version of it (blue, CUUCG).

E    Proteome-wide competition-binding assay results for the polyadenylation signal motif AAUAAA and the cleavage and polyadenylation specificity factor (CPSF) complex.

F–H    AAUAAA, UAUUUAU, and CUGUCCU motif-specific RNA binding proteins (and their complex partners) and their interaction strength to the free RNA probe; node color: pEC50; physical and functional interactions of proteins derived from STRING.

data for one positive control, the polyadenylation signal AAUAAA, are shown in Fig 6E. The protein intensities for six protein complex members of the cleavage and polyadenylation factor, which binds to the AAUAAA motif, decrease in a sigmoidal fashion in response to the concentration of the free motif-containing ligand, showing that their binding to the probe is AAUAAA-dependent and allowing for estimating EC50 and dissociation constants. Moreover, the relative intensities of these 6 members align, which is consistent with these 6 proteins to bind to the AAUAAA motif as part of a single complex. In contrast, the relative intensity of the DEAD box protein 5 DDX5 is independent of the concentration of the free motif-containing ligand indicative of unspecific binding to the AAUAAA motif, potentially to the bead itself.

In total, we identified 253 proteins which directly or indirectly (e.g., as complex members) interacted with the tested RNA motifs in a sequence-specific or sequence-independent manner. Of these, 88 proteins are annotated as RNA binding proteins (RBPDB v. 1.3.1) and 247 proteins have annotations as RNA processing or binding proteins, nuclear localization, or (mitochondrial) ribosomal proteins (according to DAVID version 6.7; Huang *et al*, 2009a,b). For further analyses, we defined sequence-specific binding proteins based on their estimated $K_d$ values. In order to be motif-specific, we required that the $K_d$ of the protein–RNA interaction be at least 10 times more potent than the next best motif or negative control. The competition-binding experiments identified 50 motif-specific interactors of the consensus polyadenylation signal sequence AAUAAA (Table EV11), 32 of which are known to bind poly(A) tails and 12 have an annotated RNA binding domain. The results unambiguously recapitulate that the motif is bound tightly by the cleavage and polyadenylation specificity factor (CPSF) complex (Mandel *et al*, 2006; Fig 6F). For the second positive control, the 3′ UTR AU-rich motif UAUUUAU (Chen & Shyu, 1995), the assay identified 38 interacting proteins (Fig 6G). These include the zinc-finger RNA binding proteins ZFP36L1 and ZFP36L2 ($K_d$ of 24 and 11 nM, respectively), which are known to destabilize several cytoplasmic AU-rich element (ARE)-containing mRNA transcripts by promoting their poly(A) tail removal or deadenylation, and hence provide a mechanism for attenuating protein synthesis (Hudson *et al*, 2004; Adachi *et al*, 2014). The competition assay also revealed interaction of many further proteins including ZCCHC11, MEX3C, MEX3D, CNBP, SKIV2, and TTC37 that are involved in mRNA decay consistent with the primary function of the AU-rich element.

The 5′ UTR motif CUGUCCU is one of the novel motifs with a predicted positive effect on median PTR ratio (1.33), which also showed a significant positive effect in the reporter assay. For this motif, we identified and quantified the interaction with 30 binding partners, including 19 proteins of the 39S mitochondrial ribosomal subunit, five proteins of the 40S ribosome complex, and four proteins with a KH domain known to be involved in splicing (Fig 6H). The ribosomal proteins bind the 5′ UTR motif tightly with an affinity of 68 nM ($\pm$31 nM std. dev.). One might speculate that the presence of this motif enhances the interaction of the 5′ UTR with the mitoribosome as well as with the small subunit of the cytoplasmic ribosome, which plays a key role in translation initiation (Aitken & Lorsch, 2012), thus leading to a higher efficiency in translation initiation. Two other proteins bound by the motif are ANGEL2, a protein known to bind the 3′ UTR of mRNAs and result

in their stabilization, and IGF2BP3, a protein which may recruit and cage target transcripts to cytoplasmic protein–RNA complexes. Like other IGF2BPs, IGF2BP3 may thereby modulate the rate and location at which target transcripts encounter the translational apparatus and shield them from endonuclease attacks or microRNA-mediated degradation (Vikesaa *et al*, 2006; Wächter *et al*, 2013). Overall, the binding partners found by this pull-down assay are ubiquitously expressed and have a positive effect on translation, which is in agreement with the sign of the estimated effect of the motif and its lack of tissue specificity.

## Discussion

Our multivariate regression analysis estimated the contribution within and across tissues of 18 sequence feature groups representing 204 post-transcriptional regulatory elements. Altogether the model predicts the PTR ratio of individual genes at a median precision of 3.2-fold from sequence alone, while the PTR ratio spans about 200-fold across 80% of the genes. For most known regulatory elements, the estimated effects were consistent with the literature, such as the effects of the secondary structures in the upstream CDS, upstream AUGs, individual nucleotides in the start and stop codon context, and *de novo* identified 3′ UTR motifs AATAAA, TATTTAT, and TGTAAATA, providing support to the functional interpretability of the model. A list of references supporting the functional evidence of sequence elements of the model on PTR ratio is provided in Table EV12. Moreover, this analysis led to the identification of novel candidate regulatory elements in 5′ UTR and 3′ UTR, whose effects are estimated to be in the range of well-known canonical motifs. Follow-up experiments provided initial functional support for these motifs. Moreover, our extended model comprising 269 additional experimentally characterized sequence features indicates that post-translational protein modifications substantially contribute to PTR ratios and would constitute an important set of features for modeling in more detail in the future.

There are limitations to this approach that should be noted. The model is additive on the logarithmic scale. However, regulatory elements likely function depending on the sequence context, the presence of other regulatory elements, and their respective distance along the transcript but also in space. Given the amount of variations across genes, such non-additive effects are very hard to be fitted. Hence, the effect of mutating a particular sequence element on a given gene may differ from the expected effect estimated by the linear model. Also, the conserved sequence elements we found associated with PTR ratio may be functional but actually play a different role because high PTR ratios correlate with other selected traits such as high mRNA levels. Experiments will help in resolving these questions. Nonetheless, our study provides interesting conserved sequence elements to follow up with mechanistic studies. We have also performed matches to the ATtRACT database, as an indication of possible RBP recognizing these motifs. ATtRACT matches, even with the highest scores, can be lenient. Also, this database suffers from the general poor charting of RBP binding sites. As a result, we shall take these matches as indicative and with caution. Another limitation is that most tissues investigated have been obtained from different donors (Wang *et al*, 2019). While it is reasonable to expect that tissue-specific effects dominate the

differential expression signal between these samples, one cannot exclude donor-specific effects as well.

Our regression approach led to a new codon metric, PTR-AI, for protein-to-mRNA ratio adaptation index, which estimates the effect of doubling the frequency of a codon in a gene on its protein-to-mRNA ratio. Using PTR-AI, codons, which inherently encode amino acids and synonymous codon usage, are the lead explanatory variable explaining about 16% of the PTR ratio variance across genes almost in every tissue we inspected. Amino acid frequency and synonymous codon usage affect PTR ratios via various mechanisms. Amino acid identity affects translation (Wilson et al, 2016; Hanson & Coller, 2018) and protein half-life (Fang et al, 2014; Zecha et al, 2018), while synonymous codon usage influences translation efficiency due to variation in the translation elongation rates of different codons (Gustafsson et al, 2012; Gardin et al, 2014; Ingolia, 2014; Yu et al, 2015; Weinberg et al, 2016; Yan et al, 2016; Hanson & Coller, 2018). Highly expressed genes contain relatively high proportions of codons recognized by abundant tRNAs with efficient codon–anticodon base-pairing. Based on this observation, several codon optimality metrics have been suggested (Sharp & Li, 1986; dos Reis et al, 2004; Pechmann & Frydman, 2013; Sabi & Tuller, 2017). However, all of these rely on some assumptions and simplifications, such as the codon adaptation index defining a set of highly expressed genes as a reference set or the tRNA adaptation index overlooking the supply and demand relationship for charged tRNAs. PTR-AI does not correlate well with codon genomic frequency or tAI adaptiveness, whereas it does correlate well with the codon decoding times estimated from several ribosome profiling datasets. Furthermore, we have shown that PTR-AI also captures the effects of amino acids on protein stability. Consequently, we suggest that PTR-AI is a more reliable codon optimality metric than previous metrics.

Our findings do not support the hypothesis of tissue-specific codon optimality. It has been suggested that there is tissue-specific codon-mediated translational control due to differential synonymous codon usage in human tissue-specific genes, which correlates with varying tRNA expression among different tissues (Plotkin et al, 2004; Dittmar et al, 2006). However, other studies found no evidence for optimization of translational efficiency by cell-type-specific codon usage in human tissues (Sémon et al, 2006; Rudolph et al, 2016). Our tissue-specific PTR-AIs, which are estimated by fitting our model separately for each tissue, do not display high variation across tissues. This result is coherent with negligible tissue-specific enrichments of expressed codons in human transcriptomes (Sémon et al, 2006; Rudolph et al, 2016), showing that tissue-specific expression is neither due to the transcription nor due to the translation of genes with particular codon contents. Further corroborating this finding, genes with high-effect codons tended both to have a high median level of protein expression and to be ubiquitously expressed. These genes were enriched for housekeeping functions. A possible explanation of these findings is that housekeeping genes have evolved for optimal coding sequence to reach high protein expression levels. Because of the ubiquitous role of housekeeping genes, their codon content in turn constrains the pool of tRNA to be rather constant across tissues. These explanations are consistent with the recent massive genomic editing experiment results, which show that codon bias of highly expressed genes maintains the efficiency of global protein translation in the cell (Frumkin et al, 2018). The lack of tissue specificity

of PTR-AIs we reported here does not contradict the differential tRNA pool regulation between proliferative and differentiating cells (Gingold et al, 2014), since our tissues are essentially constituted of non-proliferative cells.

In every tissue investigated, protein-to-mRNA ratios were higher for genes with high mRNA expression levels, leading to an approximately quadratic relationship between protein and mRNA levels across genes (Wang et al, 2019) and a larger dynamic range of expression among proteins than mRNAs. Our model partially explains this apparent amplification from sequence features, thereby showing that high protein expression levels are reached because of high mRNA levels and because of genetically encoded elements favoring the synthesis and stability of proteins. Regulatory elements that affect both the mRNA levels and protein-per-mRNA copy numbers could further contribute to this apparent amplification. Codons are known to play such a dual role since they affect translation on the one hand, and mRNA stability on the other hand. The mechanistic basis for these cross-talks between translation and mRNA stability is not fully understood. It is possible that regression approaches similar to those employed by us could help in revealing further sequence elements acting on both levels. A similar super-linear relationship had been reported before for the unicellular eukaryotes in baker's yeast (Lackner et al, 2007) and fission yeast (Csárdi et al, 2015), which appears to be absent in the prokaryote E. coli, respecting which mRNA and protein levels across genes obey a nearly linear relationship (Taniguchi et al, 2010). Prokaryotic transcription and translation are coupled processes, which do not allow post-transcriptional regulation to have an effective role in determining steady-state protein levels. In contrast, these two processes are highly uncoupled and have specialized mechanisms in eukaryotes, which are favored by the compartmentalization of eukaryotic cells. We suggest that the uncoupling of transcription and translation underlies a fundamental difference in the relationship between protein and mRNA levels across genes in eukaryotes compared to prokaryotes and may allow protein copy numbers of eukaryotic cells to span a much larger dynamic range. Further matched transcriptome and proteome datasets for a larger range of prokaryotes would help to support this model.

A comprehensive post-transcriptional regulatory code is important for interpreting regulatory genetic variations in personal genomes and in genetic engineering for biotechnological or gene therapy applications. Our study provides an important contribution by modeling codon effects, identifying novel sequence elements with potential function, and giving a framework for quantifying and assessing the role of new elements on protein-per-mRNA copy number. In the future, we expect further approaches including the analysis and integration of perturbation-based data and the mapping of post-translational regulatory elements in order to complement and refine the present analysis.

## Materials and Methods

### Protein levels, mRNA levels, and PTR ratios

The protein data in MaxQuant file "proteinGroups.txt" (see the "Data and Code Availability" section) are filtered such that the

Reverse, Only.identified.by.site, and Potential.contaminant columns are not equal to "+". Moreover, we restricted to unambiguously identified gene loci by requiring the number of Ensembl Gene IDs in the Fasta.headers column to equal 1. To calculate protein expression levels, IBAQ values equal to zero were set as missing values (NA). Next, IBAQ values were adjusted to have in each tissue the same median than the overall median by adding in the logarithmic scale a tissue-specific constant.

About 10% of the genes were reported to have 2 or more transcript isoforms in the MaxQuant file "proteinGroups.txt". We defined as major transcript isoform per gene the transcript isoform reported in the MaxQuant file "proteinGroups.txt" that had the largest sum of IBAQ values across all tissues. We used these major transcript isoforms for all tissues, to compute all sequence features and to compute mRNA levels.

For each tissue, only the mRNA replicates which had a matching protein sample were used throughout the analysis. Paired-end raw read files were quality-checked with FastQC software (Babraham Bioinformatics – FastQC A Quality Control tool for High Throughput Sequence Data, https://www.bioinformatics.babraham.ac.uk/projects/fastqc/), and the overrepresented adapter sequences were trimmed using the Trim Galore software (Babraham Bioinformatics – Trim Galore!, https://www.bioinformatics.babraham.ac.uk/projects/trim_galore/). After that, resulting read files were checked again with FastQC and the reads were mapped with STAR alignment software (Dobin *et al*, 2013) to human genome annotation Hg38.83, with the parameter of maximum number of multiple alignments allowed for a read to be equal to 1 (–outFilterMultimapNmax).

To estimate the mature mRNA levels, for each sample (each replicate in each tissue) the number of reads that map to exonic and intronic regions of the transcript (which was decided to be used based on the major protein isoform) was counted separately (Table EV2) and then normalized by the total exonic and intronic region lengths, respectively. Next, the intronic counts normalized by the intronic region length were subtracted from exonic counts normalized by the exonic region length. The resulting normalized exonic counts per sample (i.e., each replicate of each tissue) were corrected by the library size factor obtained with the Bioconductor package DESeq2 and further log-transformed ($log_{10}$). Finally, technical replicates were summarized by taking the median value. We set a cutoff of 10 reads per kilobase pair for a transcript to be treated as transcribed, which further improved the correlation between mRNA and proteins, possibly because of the poorer sensitivity of proteomics for lowly expressed genes or because of higher technical noise in low ranges of expression for RNA-Seq and for proteomics. Tissue-specific PTR ratios were computed as the logarithm in base 10 of the ratio of the normalized protein levels over the normalized mRNA levels.

### mRNA isoform level quantification

In order to obtain tissue-specific mRNA transcript isoform FPKM levels, we used Kallisto (Bray *et al*, 2016) with Gencode annotation Hg38.83 using default parameters. For each gene, the major isoform in a specific tissue was defined to be the isoform with the largest FPKM value among all isoforms with FPKM > 1. Thereon, the number of major isoforms across the 29 tissues with unique Ensembl Transcript IDs was counted per gene.

### Explained variance of protein levels and relative protein levels by mRNA levels

For protein levels, we performed a linear regression of log-transformed protein levels against either log-transformed mRNA levels of the matching tissue or the complete log-transformed mRNA levels across all tissues. Explained variance was reported as adjusted $R^2$, and statistical significance was assessed using the chi-square test for nested linear models. For relative protein levels, the same was done for log-transformed and median-centered protein levels against log-transformed and median-centered mRNA levels.

### Multi-omics factor analysis on mRNA levels and PTR ratios

Multi-omics factor analysis (MOFA; Argelaguet *et al*, 2018) was applied to mRNA levels and PTR ratio matrices (7,822 by 29) of the 7,822 genes detected expressed at the mRNA level and at the protein level in at least 15 tissues. The mRNA levels and PTR ratios were mean-centered per gene across tissues before the fitting was performed.

### Sequence features

#### 5′ UTR folding energy analysis (secondary structure proxy)
The sequence spanning 100 nt 5′ and 100 nt 3′ of the first nucleotide of the canonical start codon was extracted for all transcripts with a valid PTR value in at least one tissue. The folding energies were computed via Vienna-RNAfold package (Lorenz *et al*, 2011) with 51-nt-wide sliding window for each center position in [−75, +75] nt relative to the first nucleotide of the canonical start codon. The effect and *P*-values of the $log_2$-transformed negative minimum folding energy values at each position on median PTR across tissues were assessed individually with a linear regression model, in which all the analyzed sequence features were included as covariates. *P*-values were corrected for multiple testing using Benjamini–Hochberg correction (Benjamini & Hochberg, 1995).

#### Kozak sequence and stop codon context analysis
Linear regression was performed on every nucleotide in a [−6, +6] nt window around the canonical start and stop codons.

#### Codon frequency
Codon frequency was encoded as the $log_2$ of the frequency of each of the 61 coding codons (number of codons divided by coding sequence length). Using the frequency in natural scale led to a decreased explained variance by 1%. In addition, codon pair frequencies were modeled in the design matrix as the first 2 principal components of the codon pair frequency matrix consisting of 3,721 features.

#### Linear protein motifs
Linear protein motifs were downloaded from the ELM database (Dinkel *et al*, 2016) as regular expressions. We classified proteins as containing an ELM motif if the regular expression matched at least once in the protein sequence. Thereafter, we selected the ELM motifs significantly associating with PTR ratios in at least one tissue by utilizing LASSO feature selection (Tibshirani, 1996) where the PTR ratios were corrected for the core sequence features, which we defined as

the motifs identified *de novo*, the 5′ UTR folding energies at positions 0 and +48, start codon context, codon frequencies, codon pair bias indicators, stop codon context, UTR and CDS region lengths, PEST motifs, protein isoelectric point, and protein N-end hydrophobicity.

### N-terminal residue
The second residue of the protein sequence was extracted.

### Protein 5′ end hydrophobicity
The mean hydrophobicity value of the amino acids 2–16 at the 5′ end of the protein was calculated by the hydropathy index per amino acid values reported in Kyte and Doolittle (1982).

### Protein isoelectric point
Protein isoelectric points for 11,575 protein considered in our model were computed with the IPC-Isoelectric Point Calculator software (Kozlowski, 2016).

### PEST-region
We classified protein sequences as "PEST-region containing" if the EMBOSS program *epestfind* (Rogers *et al*, 1986; Rice *et al*, 2000) identified at least one "PEST-no-potential" hit.

### De novo *motif Identification*
Similar to Eser *et al* (2016), *de novo* motif identification was performed separately for 5′ UTR, CDS, and 3′ UTR regions by using a linear mixed model in which the effect of each individual k-mer on the median PTR ratios across tissues was assessed while controlling for the effect of the other k-mers (random effects) and region length and region GC percent (fixed effects). In order to identify k-mers which display more tissue-specific effects, the same approach was applied to tissue-specific median-centered (median being taken per gene across tissues) log-transformed PTR ratios. The model was fitted with the GEMMA software (Zhou *et al*, 2013). Motif search was executed for k-mers ranging from 3 to 8, and the *P*-values were adjusted for multiple testing with Benjamini–Hochberg's false discovery rate computed across the *P*-values of all tissues jointly. Significant motifs at FDR < 0.1 were subsequently manually assembled based on partial overlap.

### Multivariate linear model (interpretable model)

The multivariate linear model is:

$$y_{ij} = \beta_j^0 + \mathbf{x}_i^T \boldsymbol{\beta}_j + \varepsilon_{ij}, \tag{1}$$

where $y_{ij}$ is the tissue-specific PTR ratio ($\log_{10}$) of gene i and tissue j, and $\mathbf{x}_i^T$ is the $i^{th}$ row of the matrix $\mathbf{X}$ of sequence feature predictors which contains 61 features for individual codon frequencies (in $\log_2$ scale), 36 features for Kozak sequence position–nucleotide pairs, 39 features for stop-codon-context position–nucleotide pairs, three features for CDS, 5′ UTR and 3′ UTR lengths (in $\log_2$ scale), three features for CDS, 5′ UTR and 3′ UTR GC percentages, 20 features for 3′ UTR motifs, 25 features for 5′ UTR motifs (including upstream AUG), three features for CDS amino acid motifs, six features for linear protein motifs, three features for 5′ UTR folding energy, two features for codon pair bias, one feature for PEST motifs, one feature for protein isoelectric point, and one feature for

protein N-terminal hydrophobicity. The intercept $\beta_j^0$ and the vector $\boldsymbol{\beta}_j$ of the model coefficients for the $j^{th}$ tissue were estimated by ordinary least squares, i.e., minimizing the squared of the errors $\varepsilon_{ij}$.

To predict the tissue-independent effects of the sequence features, we considered the model:

$$y_{ij} = \beta_j^0 + \mathbf{x}_i^T \boldsymbol{\beta} + \varepsilon_{ij}, \tag{2}$$

where the intercepts $\beta_j^0$ varied by tissue while the coefficients of the sequence features (the vector $\boldsymbol{\beta}$) were kept equal across tissues. The intercept $\beta_j^0$ and the vector $\boldsymbol{\beta}$ of the model coefficients were estimated by ordinary least squares, i.e., minimizing the squared of the errors $\varepsilon_{ij}$.

The explained variance ($R^2$) of the PTR ratio by the sequence features was obtained by 10-fold cross-validation where in each fold the held-out data were used to have the PTR ratio predictions based on the linear regression model fit obtained from the remaining nine partitions.

### Effect of amino acids on PTR ratio
To estimate the effect of doubling the frequency of an amino acid in any gene on its $\log_{10}$ PTR ratio, we performed a modified version of the regression defined by equation 1 (for tissue-specific effects) and a modified version of the regression defined by equation 2 (general effect), whereby the amino acid $\log_2$ frequencies were considered as features instead of the codon $\log_2$ frequencies.

### PTR-AI
The tissue-specific protein-to-mRNA ratio index (tissue-specific PTR-AI) of a codon is computed as ten to the power of the estimated coefficient of the $\log_2$ frequency of this codon in the regression described by equation 1, where j is the index of the tissue of interest. It is an estimation of the fold-change on PTR ratio for a specific tissue obtained if one would double the frequency of this codon in any gene. The protein-to-mRNA ratio index (PTR-AI) of a codon is computed as ten to the power of the estimated coefficient of the codon $\log_2$ frequency in the regression described by equation 2. It is an estimation of the fold-change on PTR ratio in any tissue obtained if one would double the frequency of this codon in any gene.

### Motif analysis

### Tissue-specific motif effects
In the design matrix, all of the *de novo* identified motifs except "AUG" and "AAUAAA" are encoded as the number of motif sites in the sequence of the mRNA region (i.e., 5′ UTR, CDS, 3′ UTR). "AUG" and "AAUAAA" are encoded as binary, hence whether the motif is available in 5′ UTR and 3′ UTR regions, respectively. The tissue-specific effect of the motif is assessed by fitting all sequence features considered jointly in the linear model, with the tissue-specific PTR ratios being the response variables.

### Gene ontology enrichment
Enrichment for gene ontology categories (Ashburner *et al*, 2000) as of January 21, 2016, was performed using the Fisher exact

test and corrected for multiple testing using the Benjamini–Hochberg correction.

### Systematic motif search in RNA binding protein databases

Motif consensus sequences are searched in the RNA binding protein database ATtRACT (Giudice *et al*, 2016) by using the database Web interface at https://attract.cnic.es/searchmotif. The RNA binding protein with the highest quality score, if any, was reported as binding candidate of the motif.

### Motif 1 nucleotide mismatch logos

The sequences of each motif instance with at most 1 nucleotide mismatch were obtained from transcript mRNA sequences. The logos were created with R ggseqlogo package.

### Motif conservation analysis

Phylogenetic conservation scores for human annotation hg38 (phastConst100way from http://hgdownload.cse.ucsc.edu/goldenpath/hg38/phastCons100way), which reports conservation across 99 vertebrates aligned to the human genome, were downloaded, and the conservation scores per nucleotide were extracted for each of the motif instances without any mismatch. The significance of the enrichment scores at the motif sites compared to 10 nucleotides flanking regions was tested with a one-sided Wilcoxon test across all consensus motif occurrences in the given mRNA region (i.e., 5′ UTR or 3′ UTR).

## Codon decoding time and average amino acid decoding time

Codon decoding times for 16 human ribosome profiling datasets were obtained from RUST values (O'Connor *et al*, 2016). We estimated decoding time using the RUST ratio defined by the RUST A-site values over the RUST expected value (personal communication with Patrick O'Connor). We also included decoding times in the HEK293 cell line estimated by Dana and Tuller (2015). To estimate the average decoding times per codon, for each dataset *i* we converted the decoding times into z-scores (i.e., subtracting the mean and dividing by the standard deviation) and then used the median z-score per codon across datasets as the average normalized decoding time of the codon. The average amino acid decoding time was defined as the average codon decoding time per amino acid weighted by the codon genomic frequency.

## mRNA half-life

To estimate codon effects on mRNA half-life, for K562 cells we first called a major isoform as the highest expressed isoforms of Gencode v24 coding transcripts in the total RNA samples of Schwalb and colleagues (Schwalb *et al*, 2016) according to Kallisto (Bray *et al*, 2016). The half-life was estimated as the ratio of 5 min labeled TT-seq sample over total RNA-Seq sample (two replicates) after correcting library size with spike-in. For HeLa Tet-off cells, we used the isoforms reported by the authors (Tani *et al*, 2012), and for HEK293 cells (Schueler *et al*, 2014), we used the dominant major isoforms across the 29 tissues we have inspected. We then fitted a linear model with $\log_{10}$ mRNA half-life as response variable against $\log_2$ frequency of codons with

region length and GC content of 5′ UTR, CDS, and 3′ UTR as further covariates.

## Protein half-life

Protein half-lives for B cells, NK cells, hepatocytes, and monocytes (Mathieson *et al*, 2018) were identified only by gene name and not by isoforms, and those for HeLa cells (Zecha *et al*, 2018) by gene names and UniProt protein identifiers. We therefore mapped our transcript isoforms to these datasets by gene identifiers. We estimated the associations of the sequence features with protein half-life by multivariate regression where the response variable was the cell-type-specific log10-transformed protein half-life.

## Coding sequence 5′ end codon frequency analysis

We considered the 10,778 transcripts with CDS length greater than 460 nucleotides. Starting from the second codon, the $\log_2$ frequencies of 61 coding codons are calculated in each of the 11 non-overlapping 15-codon-long windows. For Fig EV3C, the frequency values are centered per codon across windows. In order to compare the effect of twofold codon frequency increase in the first window (codons from 2 to 16) versus the effect of the twofold codon frequency increase in the rest of the coding sequence, the codon frequencies of the whole coding sequence are replaced by the respective frequency values in the global interpretable model. The median effect of the codons across tissues is displayed in Fig EV3D.

## Explained variance of protein levels by mRNA levels and sequence features

We performed a linear regression of log-transformed protein levels against the complete log-transformed mRNA levels across all tissues and the sequence features. We also performed a linear regression of log-transformed protein levels against the complete log-transformed mRNA levels across all tissues, the sequence features, and the non-sequence features. Explained variance was reported as adjusted $R^2$.

## Non-sequence features

### m6A mRNA modification

We classified mRNAs as m6A-modified if at least one m6A peak for the same gene locus in untreated HepG2 cell line was reported in Supplementary Table 6 of Dominissini *et al* (2012).

### Protein complex membership

We classified each protein as a protein complex member if it was a subunit of at least one annotated protein complex in the CORUM (Ruepp *et al*, 2010) mammalian protein complex database (release version 02.07.2017).

### Protein post-translational modification

We downloaded protein acetylation, methylation, phosphorylation, SUMOylation, and ubiquitination data from the Phosphosite database (release version 02.05.2018) (Hornbeck *et al*, 2015) and calculated the number of modification sites per modification type for each protein. For proteins whose modification information was not

available in the downloaded dataset, we assigned 0 instead. The covariate for each of these features was defined as the $\log_2$ of the number of modifications plus 1 (pseudocount).

### RNA binding protein targets

We classified transcripts as targets of 112 RBPs if they contained at least one peak in the eCLIP dataset of Van Nostrand et al (2016) as processed earlier (Avsec et al, 2018).

### RNA binding protein across-tissue covariation with target genes:

Among 112 RBPs whose binding evidences were used in our integrated model, 64 of them were expressed in at least 15 tissues with an mRNA level standard deviation across tissues > 0.1. In order to see across-tissue expression covariation between these RBPs and their target genes, for each RBP we calculated the Spearman's rho between its protein level expression and mRNA levels in other genes (again expressed in at least 15 tissues and with mRNA ratio standard deviation > 0.1). The significance of the correlation coefficient distribution difference between target and non-target genes was assessed with two-sided Wilcoxon test.

### miRNA targets

Many miRNAs in the miRTarBase database (Chou et al, 2018) have very few reported targets, leading to no improvement explained variance. Therefore, we filtered for the miRNAs which have at least 200 experimentally validated target genes in our dataset and classified the genes accordingly as targets for these miRNAs. Due to high collinearity between binding evidences of different miRNAs, we applied PCA to the $11,575 \times 296$ binding evidence matrix and selected as features the 150 first principal components that explained 95% of the variance in the target genes of 296 miRNAs.

### Independently matched transcriptome–proteome dataset

We used data from Kremer et al (2017). As originally reported, these data showed strong technical effects. To be on the safe side, we restricted the analysis to six samples (sample IDs: #65126, #73804, #78661, #80248, #80254, and #81273) that belonged to the same cluster.

### Validation of RNA motifs using a GLuc/SEAP reporter assay

We assayed the expression of one reporter gene on a plasmid (Gaussia luciferase, GLuc) as a function of the presence of a motif, while the second, constitutively expressed reporter gene (secreted alkaline phosphatase, SEAP) was used as internal control for variation in transfection efficiency and plasmid number. The pEZX-GA01 vector and the pEZX-GA02 vector (GeneCopoeia), containing Gaussia luciferase (GLuc) as a reporter and a constitutively expressed secreted alkaline phosphatase (SEAP) as an internal control, served as basic vectors of our 5′ UTR and 3′ UTR constructs, respectively. We cloned the SV40 promoter and the 5′ UTR motifs upstream of Gaussia luciferase ORF between the EcoRI and the XhoI sites of the pEZX-GA01 vector and 3′ UTR downstream of the Gaussia luciferase stop codon between the EcoRI and the XhoI sites of the plasmid pEZX-GA02. The list of the motifs and controls used in the study is available in Table EV8. The luciferase assay was performed using Secrete-Pair™ Dual Luminescence Assay Kit (GeneCopoeia). A total of 100,000 HEK293-FT cells per construct were plated in 12-well plates. The following day, cells were transfected with 1 μg of DNA of each construct using Lipofectamine 2000 transfection reagent (Life Technologies) according to the manufacturer's protocol. The medium was changed 24 h after transfection, and the cell culture medium was collected 48 h after transfection. GLuc and SEAP activities were measured with the Secrete-Pair™ Dual Luminescence Assay Kit (GeneCopoeia) according to the manufacturer's protocol on Cytation3 imaging reader (BioTek). Each construct was measured in three technical and three biological replicates in two independent experiments with intensity measurements collected on at 5 time points (0, 2, 4, 7, and 10 min).

For each motif separately, we assessed the per-experiment significance of the effect of the motif versus its scrambled counterpart with a two-level nested ANOVA model fitted as a linear mixed model in which the response is the log-intensity ratio of GLuc over SEAP. In each of these motif-specific models, the motif type (motif versus scrambled motif) is treated as the fixed effect while the replicate identifier is treated as the random effect (Table EV9, Appendix Figs S15 and S16). Thereafter, we combined the P-values of the replicate experiments using Fisher's method (Fisher, 1925) and corrected for multiple testing with Benjamini–Hochberg correction (Benjamini & Hochberg, 1995).

### Competition-binding assay to identify RNA motif-binding proteins

The experiments were performed in three biological replicates and in two independent experiments. HEK293-FT cells were grown in DMEM medium supplemented with 10% (v/v) FBS, 1% (w/v) non-essential amino acids, 1% (w/v) L-glutamine, and 1% (w/v) G418 (Geneticin; Thermo). Confluent cells were harvested by mechanical detachment followed by centrifugation and washing with cold Dulbecco's phosphate-buffered saline containing $Ca^{2+}$ and $Mg^{2+}$. Cell extraction and preparation of the lysate for the competition-binding assay were performed as described (Médard et al, 2015). The preparation of RNA-beads for affinity purification of RNA binding proteins was performed as follows: NHS-Sepharose beads (Amersham Biosciences) were washed with DMSO ($4 \times 10$ ml/ml beads) and reacted with RNA oligos with 5′ amino modifier C6 (Table EV11) (50 nmol/ml beads; Integrated DNA Technologies, Inc.) for 20 h on an end-over-end shaker in the dark in the presence of triethylamine (30 μl/ml beads) in DMSO and $H_2O$ (1.8 vol of DMSO and 0.2 vol ddH$_2$O for 1 vol of beads). Next, aminoethanol (50 μl/ml beads) was added and the mixture was kept shaking for an extra 20 h in the dark. The beads were washed with DMSO (10 ml/ml beads) and ethanol ($3 \times 10$ ml/ml beads) and stored in ethanol (1 ml/ml beads) at 4°C.

The competition-binding assay itself, the quantitative label-free LC-MS/MS analysis, and curve fitting were also developed according to Médard et al (2015). Briefly, the diluted cell lysates (2.5 mg of total proteins/well) were incubated for 1 h at 4°C in an end-over-end shaker with 0 nM (water control), 0.3, 1, 3, 10, 30, 100, 300 nM, 1, 3, 10, and 30 μM of the free RNA oligos dissolved in RNase-free water. The preincubation step was followed by incubation with 20 μl settled beads for 30 min at 4°C. The water control lysate was recovered and incubated similarly with RNA oligo beads

as a pull-down of pull-down experiment to calculate the depletion factor. The bound proteins were subsequently eluted with 60 μl of 2× NuPAGE LDS sample buffer (Invitrogen, Germany) containing 50 mM DTT. Eluates were alkylated with CAA, and in-gel digestion was performed. The resulting peptides were measure using nanoflow LC-MS/MS by directly coupling a nanoLC-Ultra 1D+ (Eksigent) to an Orbitrap Elite mass spectrometer (Thermo Fisher Scientific). Peptides were delivered to a trap column (75 μm × 2 cm, self-packed with Reprosil-Pur C18 ODS-3 5 μm resin; Dr. Maisch, Ammerbuch) at a flow rate of 5 μl/min in solvent A (0.1% formic acid in water). Peptides were separated on an analytical column (75 μm × 40 cm, self-packed with Reprosil-Gold C18, 3 μm resin; Dr. Maisch, Ammerbuch) using a 100-min linear gradient from 4 to 32% solvent B (0.1% formic acid, 5% DMSO in acetonitrile) in solvent $A_1$ (0.1% formic acid, 5% DMSO in water) at a flow rate of 300 nl/min (Hahne *et al*, 2013). Full scans (m/z 360–1,300) were acquired at a resolution of 30,000 in the Orbitrap using an AGC target value of 1e6 and maximum injection time of 100 ms. Tandem mass spectra were generated for up to 15 peptide precursors. These peptide precursors were selected for fragmentation by higher energy collision-induced dissociation (HCD) using 30% normalized collision energy (NCE) and analyzed in the Orbitrap at a resolution of 7,500 resolution using AGC value of 2e5 and maximum injection time of 100 ms. For peptide and protein identification and label-free quantification, the MaxQuant suite of tools version 1.5.3.30 was used. The spectra were searched against the UniProt human proteome database with carbamidomethyl (C) specified as a fixed modification. Oxidation (M) and Acetylation (Protein N-Term) were considered as variable modifications. Trypsin/P was specified as the proteolytic enzyme with two maximum missed cleavages. Label-free quantification (Cox *et al*, 2014) and the match between runs function was enabled. The FDR was set to 1% at both PSM and protein level. Protein intensities were normalized to the respective water control, and IC50 and EC50 values were deduced by a four-parameter log-logistic regression using an internal pipeline that utilizes the "drc" package (Ritz *et al*, 2015). A $K_d$ was calculated by multiplying the estimated EC50 with a protein-dependent correction factor (depletion factor) as previously described (Médard *et al*, 2015).

## Data and Code Availability

Transcriptome sequencing and quantification data are available at ArrayExpress under accession ID E-MTAB-2836 (www.ebi.ac.uk/arrayexpress/experiments/E-MTAB-2836/). The raw mass spectrometric data and the MaxQuant result files are available from the PRIDE database under accession number PXD010153 for the pull-down dataset and PXD010154 for the tissue profiling dataset (https://www.ebi.ac.uk/pride/archive/projects/PXD010153 and https://www.ebi.ac.uk/pride/archive/projects/PXD010154).

Analysis scripts are available at https://github.com/EraslanBas/HumanTransProt.

**Expanded View** for this article is available online.

## Acknowledgements
We thank Stephanie Heinzlmeir for valuable discussions on the RNA-motif competition-binding assay. We thank Jun Cheng for his help on the RNA half-life estimations. We thank Terence Hwa for discussion on prokaryotic transcriptome and proteome relationship. This work was in part funded by the German Excellence Initiative cluster Center for Integrated Protein Analysis Munich (CIPSM). Dongxue Wang is grateful for a scholarship from the Chinese Research Council. Basak Eraslan has been funded by a fellowship through the Graduate School of Quantitative Biosciences Munich (QBM). Holger Prokisch, Basak Eraslan, and Julien Gagneur have been funded by European Union's Horizon 2020 research and innovation program under grant agreement No. 633974.

## Author contributions
MU, HP, HH, BK, and JG conceived and designed the study. DW, MG, and AA performed experiments. BE, DW, TW, HH, BK, and JG conceived the data analysis. BE, BH, TW, TH, and JG analyzed the data. BE, DW, MG, HH, BK, JG, and TW wrote the manuscript. FP contributed samples and data to the project.

## Conflict of interest
HH and TH are employees of OmicScouts GmbH. HH and BK are co-founders and shareholders of OmicScouts GmbH. BK has no operational role in OmicScouts GmbH.

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
