## [Review Process File · Molecular Systems Biology]

Quantification and discovery of sequence determinants of protein per mRNA amount in 29 human tissues

Basak Eraslan, Dongxue Wang, Mirjana Gusic, Holger Prokisch, Björn Hallström, Mathias Uhlen, Anna Asplund, Frederik Ponten, Thomas Wieland, Thomas Hopf, Hannes Hahne, Bernhard Kuster, Julien Gagneur

Review timeline:	Submission date:	21 June 2018
	Editorial Decision:	10 August 2018
	Revision received:	6 December 2018
	Editorial Decision:	8 January 2019
	Revision received:	22 January 2019
	Accepted:	23 January 2019

Editor: Maria Polychronidou

Transaction Report:

1st Editorial Decision

10 August 2018

Thank you again for submitting your work to Molecular Systems Biology. We have now heard back from the three referees who agreed to evaluate your study. As you will see below, the reviewers think that the presented findings are potentially interesting. They raise however a series of concerns, which we would ask you to address in a major revision.

Without repeating all the points listed below, some of the more fundamental issues are the following:

- Reviewer #1 suggests further analyses of RNA-binding motifs and RNA-binding proteins to provide potential explanations for PTR control.
- As reviewers #1 and #3 point out, several statements throughout the manuscript need to be modified in order to avoid overstatements regarding causality (e.g. between sequence features and PTR ratios).
- Reviewer #2 provides several constructive suggestions on how to make better use of the extensive data available in order to enhance the impact of the study.
- The reuse of figure panels that are also presented in the related manuscript MSB-18-8503 should be avoided. Please refer to the other manuscript instead.

All other issues raised by the reviewers need to be satisfactorily addressed. As you may already know, our editorial policy allows in principle a single round of major revision so it is essential to provide responses to the reviewers' comments that are as complete as possible. Please feel free to contact me in case you would like to discuss in further detail any of the issues raised by the reviewers.

REFeree REPORTS

Reviewer #1:

Summary

In this study, the authors analyzed previously measured matched transcriptomics and proteomics data for about 11,500 protein across 29 healthy human tissues, with an average of 3,600 quantified proteins and transcripts per tissue. The analysis centers around the relationship of protein abundance to transcript abundance ratios (PTR) and their relationship to a larger number of sequence features. These features are investigated individually for their predictive power using linear regression and features with <10% FDR are selected. This resulted in a final multivariate linear model with 175 different sequence features (incl. codons frequency, nucleotide and amino acid motifs in 5', CDS, 3' regions and calculated properties such as hydrophobicity) to predict between gene PTRs with an average across tissues of 22% of the PTR variance explained. The selected features contain motifs known to be related to PTR as well as newly discovered ones. An independent previously published dataset evaluated for the selected features showed consistent effects, suggesting that these findings are not dataset specific.

General remarks

While I can see the merit of the data set used (which seems to not be primary published in this article, but not information on the source of the data is given) and also the principle question of difference of protein and transcript abundance and their relationship, the manuscript seems to lack an important finding or conclusion and instead descriptively iterates numerous sequences features. Importantly, the correlation of these features with the PTR does not imply causality. Except for the reporter assays, which appear to confirm a weak contribution of some of the tested motifs, such causal relationships remain untested: The work is mainly an analysis of sequence features correlating with the PTR, not controlling the PTR (as claimed in the abstract and throughout the paper).

On a more technical side, it is unclear how the authors apply Benjamini-Hochberg multiple testing correction. This can make a difference between a 10% or 95% error rate. Furthermore, no model selection criteria taking into account model complexity like AIC/BIC is used. As such it is not possible to assess the relevance of the new features detected. Also, the manuscript presents through various smaller things like cut graphics in figures, and unreferenced figure panels, unclear sentences a certain lack of attention to detail, which does not increase confidence.

The Wang et al (MSB-18-8503) manuscript in contrast to the Eraslan et al (MSB-18-8513) manuscript is well written, has high quality figures & supporting figures, etc. The Wang/8503 manuscript is publishing the generated data incl. the methods description and as such presents a classical data resource paper with a sound proteogenomics analysis but limited findings and conclusions. In contrast, the Eraslan/8513 manuscript is not only not a resource manuscript, but also presents a way less sound analysis, to the point where it's reliability is unclear, and even if true limited finding and impact.

Fig. 1A,B,C in MSB-18-8513 are basically equal to Fig. 1A, 2A, and 2B of MSB-18-8503, with the ones in MSB-18-8503 being generally better. Also note that for the same data 1C and MSB-18-8503 2B reported different regression slopes 2.76 and 2.6 respectively.

Major points

1. Overall, the authors should be more carefully distinguish between correlation and causality. For example, in the abstract they refer to their work as a "catalogue of sequencing features controlling protein-to-mRNA (PTR) ratio". However, the observation that a given feature correlates with the PTR does not imply a causal relationship. More generally, let's assume A correlates with B. Let's further assume that this correlation is not random (see above for the possible FDR issue in this study). This finding could be interpreted as (i) A causes B, (ii) B causes A or (iii) A and B both

correlate with the confounding factor C, which causes a spurious association between A and B. A fundamental problem throughout this paper is that only the first possibility (i.e. that a feature causes increased/decreased PTR) is considered while the third possibility (spurious correlation) is in my opinion more likely.

As an example let's focus on the codon bias. In many organisms there is a significant positive correlation between gene expression level and the degree of its codon bias (for a good review see Plotkin and Kudla, Nature Reviews Genetics, 2011). Given that the PTR positively correlates with gene expression, it is expected that codon bias also correlates with the PTR. Importantly, this does not imply a causal relationship: In fact, there is "an overabundance of plausible explanatory models" for this observation (Plotkin and Kudla, 2011). The idea that "cognate tRNA concentrations influence decoding time" (page 9) is just one of many plausible explanations.

2. Related to point 1 above, it is remarkable that almost all sequence features investigated correlated with the PTR in most of the tissues. If these features would indeed control the PTR via some mechanism, the corresponding trans-acting factors recognizing them (e.g. RNA-binding proteins, microRNAs) would have to be expressed in all tissues. I think it is more likely that observed correlations are caused by confounding factors that do not differ between tissues.

3. I am missing a systematic comparison of kmers found in this study with known RNA-binding motifs (such as in Ray et al., Nature, 2013). Such analyses might provide testable hypotheses on the molecular mechanisms regulating the PTR. Along the same lines, it might be informative to specifically look at the few RBPs (and their motifs) that are known to be expressed in a tissue-specific manner.

4. The authors apply a Benjamini-Hochberg multiple testing correction to the p-values when evaluating individual features. It remains however, unclear if this correction is only done for each group of feature e.g. 5' UTR sequence features, or across all features tested.

5. Is this testing correction done per tissue or across tissues? Since features are selected if they are below 10% error rate in at least one tissue, multiple testing correction needs to be across tissues, otherwise the true error rate will be a lot higher, namely 95% for 29 tissues!

6. In modelling the significance of feature and overall quality are not the only relevant criteria. The model complexity is crucial as well to demonstrate the importance of features. As such it is unclear why the author do not employ model selection criteria taking into account the number of features such as Akaike or Bayesian information criterion (AIC or BIC).

Minor points

7. Protein sequence features are mostly (i.e. with the exception of hydrophobicity) given without significance statement. Are these 10% FDR significant?

8. Why was the ELM motifs assessed with the LASSO feature selection method, rather than the otherwise applied linear regression based assessment?

9. Methods: the origin of the proteomics proteinGroups.txt is not given.

10. Methods: "To estimate the mature mRNA levels, the exonic and intronic FPKM values were calculated separately and then the normalized intronic levels were subtracted from the normalized exonic levels." - From which files are FPKM values calculated and how? How were the intronic/exonic levels normalized?

11. "Requiring at least 10 sequencing-depth normalized reads per kilobase pair further improved the correlation between mRNA and proteins, likely because of the poorer sensitivity of proteomics for lowly expressed genes." and "We set a cut-off of 10 reads per kilo base pair for a transcript to be treated as transcribed, which further improved the correlation between mRNA and proteins, likely because of the poorer sensitivity of proteomics for lowly expressed genes." - For both the mention another similarly likely reason, which should be discussed here, for improved correlation is the higher noise level of mRNA abundance calls in lower coverage RNAseq regions.

12. "[...] the relative effects of codons in this region were highly correlating with the effects of the rest of the coding region (Materials and Methods, Spearman's $\rho=0.38$, $P < 0.0024$, Fig EV5C)." - $\rho=0.38$ does not qualify as highly correlated

13. Fig. 1C - There are no visible data point with Brain protein levels below $1e+03$, y-axis should be the most go down to $1e+01$ not 0, or both axis should be the same. The current axis make plot look weird. Also, units (e.g. FPKM) of axis are missing.

14. Fig. 3C - What is the UUCGCC box?

15. Fig. 6B - The colors of the datasets are hardly visible; e.g. reduce line thickness of mean/median line.

16. Fig. 6E,F,G,H - These panels are not referenced in the main text at all!

17. These graphics are not publication quality and show lack of attention to detail:

i. Fig. 1D - Sloppy x-axis labelling

ii. Fig. 2B - Icon human has lines with truncated unreadable labels.

18. "We used as protein expression levels the per tissue median centered \log_{10} IBAQ values, settings missing values (NA) those IBAQ values equal to zero." - Please fix the language in this sentence

19. References are not formatted consistently nor according to journal specifications

Reviewer #2:

Eraslan present a in-depth analysis of sequence and experimental features of the RNA and protein data collected in the accompanying manuscript by Wang et al, quantifying expression for >11,000 genes across 29 human tissues. The work focusses on insights gained from protein-to-RNA (PTR) ratios and presents a detailed extension of work published in 2010 on one cell line.

The analysis is thorough and well-presented, but in my view, could move even more beyond the 2010 paper and make use of the new, much more extensive data available. I am excited about the work, but an only support it fully once the criticisms below have been addressed.

MAJOR

Codon optimality.

Even in the abstract the authors mention a new metric of codon optimality, but I fail to find it. The way it is written at the moment, it suggests the authors developed a new measure that helps us understand codon usage in multi-cellular organisms. I find this part of the analysis/results very interesting and important, but its presentation is entirely suboptimal.

I think the authors mean the ribosome dwell-time. That is not a new measure of codon optimality, that is just what it is: ribosome dwell-time. I would expand the discussion of what that means for translation (initiation vs. elongation etc etc), but not claim the authors developed a new measure (unless I failed to find it).

Further, it is very interesting to see this for multi-cellular organisms. Most models have analyzed single-cellular organisms. This should be looked at in more depth and/or discussed.

p. 16 middle - of course many of the codon optimality measures break down for multi-cellular organisms.

p. 16 end of second-but-last paragraph - analyzing codon optimality with PTR is NOT a new measure of codon optimality. That is entirely misleading. And also not so new. See above, dwell time is much more interesting.

p. 16 bottom - codon optimality does not vary much across tissues. The authors discuss this in passing, somewhat hiding a 'negative' result. Please step back from this and rework this discussion to realize that this 'negative' result is actually the interesting part: tissue-specific gene expression

levels seems to be more set at the RNA than at the post-transcriptional level, as PTRs do not vary much across tissues and codon dwell time doesn't. Is that the right interpretation? This would go along with other discussions (e.g. Vogel, Marcotte Nat Rev Gen 2012 and others) on post-transcriptional regulation fine-tuning whatever has been set by RNA level... I think there is potential for some more interesting, more novel insights/discussion.

Coupling of translation and RNA degradation.

The discussion should contain references for other work highlighting this (it exists, e.g. Sun/Cramer).

Tissue-specificity.

Unfortunately, the authors miss out entirely on the enormous benefit their dataset provides: tissue-specific expression of RNA and protein. Of course the sequence information is identical, but still, there are many questions that can be addressed. And even if the data isn't good or large enough to have the resolution, data grouping or at least a discussion of trends/limitations would be novel. Otherwise, it is just an 'update' of the 2010 work. Please don't miss out on a chance to move beyond that.

Example questions would include:

- For brain and testis, with many tissue specific genes, are there trends that are different than in other tissues?
- Some of the RBP binding motifs etc that have been found, are they enriched in tissue-specific genes or in genes with particularly high/low PTRs in different tissues?
- The new/missing proteins discussed in the other paper, do they have particular characteristics?
- How does the overall model vary across tissue types? If it doesn't (low resolution), can you take the extremes, or group the data, or just compare high vs low PTR ?
- What is the significance of Fat being different in Figure 2D?
- Any tissue specific effects in Figure 4?

Transcription vs. post transcriptional regulation.

Is there any way one can tie in how much is explained by just looking at RNA levels? And how much by PTR ratios (i.e. post-transcriptional regulation)? It has been done before, but a quick reference to how the new data compares would be nice.

RBP analysis.

I think this part is very interesting and also nice to see the experimental follow-up. This can be highlighted more. Also, p. 14 bottom, the authors discuss all those data but then leave the reader hanging with an interpretation or further discussion. What do these proteins do? How does this explain the trends? Do they show tissue-specific expression? Would that explain tissue-specific effects? This is a nice example to get some more discussion of tissue-specific characteristics of the data into the work, which would increase its novelty and also - by highlighting some examples in more depth, makes it much more accessible to a broader readership.

The Van Nostrand data (2016) etc is still so recent that its in-depth connection to the tissue-specific data here (RBP expression, target expression etc) would be a unique chance to increase the impact of the paper.

Then this should be mentioned in the abstract.

MINOR

1. Figure 1C: it appears that low-abundance, noisy RNA has not been filtered (on the left). Perhaps at least discuss the impact of this?
2. I do not understand legend/figure in Figure 3C

Reviewer #3:

The manuscript "Quantification and discovery of sequence determinants of protein per mRNA amount in 29 human tissues" by Eraslan et al uses regression analysis to study the impact of various sequence features on protein-to-mRNA (PTR) ratios. This is essentially a follow-up on an earlier report in MSB (Vogel et al, 2010), but the improvement in data size and quality is substantial enough to warrant a repetition of the pioneering study. In addition, to my knowledge Eraslan et al

are the first to use such data to search for - and identify - novel sequence motifs that may control PTR ratios. As such I believe this manuscript is a valuable addition to our current understanding of mRNA and protein expression control and provides a blueprint for similar association studies in the future.

There are a few points that need to be addressed:

λ Overlap with accompanying paper: Although the manuscripts by Wang et al and Eraslan et al can stand on their own, there is at present still some unnecessary overlap. The conclusion that there is a quadratic relationship between mRNA and protein levels in every tissue should not be presented in both papers, particularly because it only confirms earlier reports by the authors and other groups. For example, Fig. 1C in Eraslan et al is almost identical to Fig. 2B in Wang et al. The authors need to decide which of the two reports will include that finding, and the other paper should then refer to this rather than showing nearly identical material.

λ Transcript isoforms: The authors select the major transcript isoform for each gene based on its average protein abundance across all tissues. I think they need to justify that decision and provide some statistics to show if that is a sensible approach. For example, are the most abundant protein isoforms generally produced by the most abundant transcripts? Are there any large transcript isoform differences across the tissues, i.e. is it reasonable to use one transcript isoform per gene across all tissues?

λ Dynamic range: The dynamic range of proteins shown in this paper is 1-2 orders of magnitude lower than in the accompanying paper. Why is that? Perhaps because the subset of proteins included here are enriched for housekeeping functions? It would be good to show that, because gene regulation of constitutively expressed and developmentally induced (tissue specific) genes can be quite different. This would have implications for their overall conclusions. In Fig 1B, I also find it confusing to show two completely different units (FPKMs and iBAQs) with the same x axis scale.

λ Perhaps I missed this, but it would be interesting to know how much protein-to-mRNA ratios actually vary between tissues. I suppose there are some genes where the ratio varies more and others that vary less, so that may be interesting to point out.

λ Out-of-frame AUGs in the 5' UTR decrease protein-to-mRNA ratios. This is an interesting observation. Producing a non-functional or unstable protein seems like quite a wasteful way of reducing a gene's PTR, so I'm wondering how many out-of-frame AUGs we are talking about here? Are out-of-frame AUGs in general depleted from 5' UTRs, as one might expect? In other words, are these cases where gene expression has yet to be optimised more completely in evolutionary terms? Or could it be that such out-of-frame translation gives rise to stable but unannotated gene products?

λ The novel 5 k-mers: With an FDR of 10% there is a good chance that one of the 5 motifs is a false positive, so I think more data are needed to allow the reader to get a better idea of how likely each of these motifs are to have an effect. For example, the motif CUGUCCU is found in 323 genes. When you say it increases PTR by 33%, is that supported by all these genes or are there perhaps a few genes with this motif that have a dramatically increased PTR, thus driving the average impact of the motif? Likewise, the fact that these genes are enriched for certain GO terms really doesn't mean much without reporting an actual fold-enrichment, or percent of genes associated with those GO terms and p-values etc. Finally, the evolutionary conservation is quite critical here. It looks visually convincing for three motifs but not for CUGUCCU and GGCGCCCG. It would be good to have some statistical test (rather than just the sequence plot) to support the fact that these motifs are evolutionary conserved in a statistically significant manner.

λ Codon frequency vs amino-acid frequency: The authors claim that codon frequency is the single most important sequence feature to predict PTR ratios, but I'm not sure their data fully support this conclusion. In Fig 3C it appears that the PTR ratios depend mainly on the amino acid frequency, as most codons that encode for the same amino acid show the same directionality of PTR changes. Moreover, the authors state the predictive power of codon frequency is only 2% higher than that of amino acid frequency (17% vs 15%), which seems to me would support the conclusion that the latter is more important, since codon frequency will reflect amino acid frequency more strongly than the other way around. Perhaps the authors can dissect this issue a bit further. For example, how important are codon vs aa frequency when you distinguish between amino acids encoded by multiple codons vs single codons? Can you find a way to better distinguish between these two effects? I think this is quite important since the biological mechanisms behind them will be quite different - codon usage points to an impact of the gene expression programme (availability of tRNAs etc), while aa frequency points to biological properties of the proteins themselves (protein stability or function). Both effects will probably play some role but the relative contribution of each would be important to estimate. Especially because codon usage (or aa frequency) is by far the most

important feature for PTR prediction in the multivariate regression.

λ De novo motifs when controlling for codon pair frequencies (last paragraph of "Codon usage" section): Why were these motifs not picked up in the earlier motif search? Given that there is no evidence for these motifs to be conserved and that they are found in rather selective frames / positions, I would say this is too speculative to be included without additional support.

λ Protein sequence features: I think there is a danger here to confuse cause with correlation. For example, just because nuclear localisation signals correlate with lower PTR ratios does not mean that the signal peptide sequence itself is mechanistically involved in it. Isn't it more likely that nuclear proteins per se might have lower PTR ratios? That could simply be driven by a higher proteasome activity in the nucleus. For example, I believe proteins involved in nuclear RNA pre-processing are generally less stable than cytoplasmic proteins. In other words, what if the sequence motifs you pick up are simply an indirect measure of protein function? I guess it's impossible to really address that without biological follow-ups (as you perform for one motif later), but maybe you could address this point by associating GO terms with PTR ratios. If nuclear proteins have a lower PTR ratio in general, then it is unlikely that the NLS contributes directly to it. I understand the goal is to predict PTR ratios from gene sequence, but while some of the features you describe are directly related to the mechanisms of translation / degradation, the impact of others may be very indirect.

λ Variance explained across tissues: With the multivariate regression models, where you combine the individual features to predict PTRs, I think you need to be careful with the phrasing. The phrase "This model explained 22% of the variance in median across tissues" to me suggests that you explained the difference in PTRs of one gene across many tissues, whereas I believe you are explaining the variance of PTRs between genes, showing the distribution of those predictions across the 29 tissues as boxplots. This should be formulated a bit more carefully to avoid confusion.

λ Given the central importance that the authors attribute to the value 3.2 this deserves a representation in a figure. The currently closest panel is Fig 5B. A light reader could draw from this the impression that the model predicts a PTR ratio to be about $1e+05$.

λ The conclusion that "The predicted PTR ratios positively correlated with the mRNA levels (Fig 5D)." should probably not be drawn without a reference to the very broad spread also visible in that figure.

λ The speculation "We suggest that the uncoupling of transcription and translation underlies a fundamental difference in the relationship between protein and mRNA levels across genes in eukaryotes compared to prokaryotes and may allow protein copy numbers of eukaryotic cells to span a much larger dynamic range." falls short in recognising evolutionary considerations as voiced by the Hurst lab, for example, and backed in vertebrates by recent work of the Rappsilber lab.

λ Figure legends could help readers better in understanding the displayed data, for example by spelling out acronyms (e.g. LIG_KEPE_1 and others used in Figure 4 are not intuitively understood by me).

1st Revision - authors' response

6 December 2018

Reviewer #1:

Summary

In this study, the authors analyzed previously measured matched transcriptomics and proteomics data for about 11,500 protein across 29 healthy human tissues, with an average of 3,600 quantified proteins and transcripts per tissue.

Response: This is not exact. In page 5 we wrote as "an average of 3,603 (31%) PTR ratios per tissue could not be quantified and were subsequently considered as missing values.". The median of quantified PTR ratios is 7,972 (min 7,300, max 8,869). We re-wrote the sentence at the end of the first result paragraph to avoid confusions.

R1: The analysis centers around the relationship of protein abundance to transcript abundance ratios (PTR) and their relationship to a larger number of sequence features. These features are investigated individually for their predictive power using linear regression and features with <10% FDR are selected. This resulted in a final multivariate linear model with 175 different sequence features (incl. codons frequency, nucleotide and

amino acid motifs in 5', CDS, 3' regions and calculated properties such as hydrophobicity) to predict between gene PTRs with an average across tissues of 22% of the PTR variance explained. The selected features contain motifs known to be related to PTR as well as newly discovered ones. An independent previously published dataset evaluated for the selected features showed consistent effects, suggesting that these findings are not dataset specific.

General remarks

While I can see the merit of the data set used (which seems to not be primary published in this article, but not information on the source of the data is given) and also the principle question of difference of protein and transcript abundance and their relationship, the manuscript seems to lack an important finding or conclusion and instead descriptively iterates numerous sequences features.

Response: The data resource paper has been submitted back to back. It is the manuscript Wang et al (MSB-18-8503, also on biorxiv as <https://doi.org/10.1101/357137>) that we referred to as "accompanying manuscript"), which this reviewer had access to and refers to later on. There is also a Data availability section in Materials and Methods. We have now added a reference to it after the first occurrence of the filename "proteinGroups.txt" (see detail point below).

Regarding the importance of the findings and the approach, we are discovering two novel functional (which we validate) regulatory motifs controlling protein to RNA ratios in the human genome using a comprehensive integrative approach. Reviewer #3 instead considers our work an "addition to our current understanding of mRNA and protein expression control and provides a blueprint for similar association studies in the future." With the large amount of mechanistic studies whose results are scattered across primary research papers and reviews on the one hand, and with the availability of large scale multi-omics dataset on the other hand, there is a need for approaches like ours that aim at assessing the importance of known elements quantitatively across multiple conditions where they have not been yet studied, and provide testable hypotheses for candidate novel regulatory elements.

R1: Importantly, the correlation of these features with the PTR does not imply causality. Except for the reporter assays, which appear to confirm a weak contribution of some of the tested motifs, such causal relationships remain untested: The work is mainly an analysis of sequence features correlating with the PTR, not controlling the PTR (as claimed in the abstract and throughout the paper).

Response : We agree with the reviewer that accurate and precise language is important. In order to avoid any misunderstanding in our revised manuscript, we have systematically searched for ambiguities in our wording. We identified three issues: a) one heading, b) the use of the word 'effect' in a statistical sense, and c) the use of the word 'quantify'.

To a) The results heading "Sequence determinants of protein-to-mRNA ratio" was indeed wrong and now reads as follows "Sequence features predictive of protein-to-mRNA ratio".

To b) In the regression context, the term "effect" is to be understood in the statistical sense, as the predicted effect of the linear regression model (as in "effect size", "random effect model", "mixed effect model", etc). We now stress further the difference between the estimated from observational data and the mechanistic effect.

We think this clarifies the usage of the word 'effect'. We could also have "associated effect" in place effect of "effect" everywhere if wished. The section in question now reads as follows: "The effects of each sequence feature on PTR log-ratio are estimated using the joint model, thereby controlling for the additive contribution of all other sequence features (Dataset EV6). We underscore that these reported effects are estimates from the multivariate model from observational data. Hence, they may or not reflect the effects of creating or removing a single sequence feature in a given gene because they are estimated from observational data and because of regression artefacts such as spurious correlations and regression-toward-the-mean, and because of the simplifying modeling assumption that the regulatory elements function independently of each other."

To c) We have now replaced “quantify” with “estimate” when referring to effects estimated by the model.

We would like to emphasize that the original manuscript systematically used the word “associating” and was already clearly separating the modeling from the functional validation part. Also, the abstract was not claiming that our elements are causal. The following sentence of the abstract describes the motivation for the study, not a claim of the results: “Despite their importance in determining protein abundance, a comprehensive catalogue of sequence features controlling protein-to-mRNA (PTR) ratios and a quantification of their effects is still lacking.”

We have re-phrased the sentence “We analyzed the contribution of known sequence determinants of protein synthesis and degradation and 15 novel mRNA and protein sequence motifs that we found by association testing.” as:

“We estimated by regression analysis the contribution of known sequence determinants of protein synthesis and degradation and 45 novel mRNA and 3 protein sequence motifs that we found by association testing.” (we report more motifs as we have now looked into tissue-specific effects).

Note that the next abstract sentence made clear that only two of the novel elements found by association have been validated:

“A reporter assay provided significant functional support for two novel UTR motifs and a proteome-wide competition-binding assay identified motif-specific bound proteins for one motif.”

Hence, while we agree that there was some possible confusion and one clearly misleading heading, we disagree with the notion that we were claiming causality throughout the paper and abstract.

Finally, we are now providing new table (Table EV12) that lists what sequence features are causal and those for which we only have correlative evidences. We think this will guide the readers with respect to the interpretation of the sequence features and their evidence.

R1: On a more technical side, it is unclear how the authors apply Benjamini-Hochberg multiple testing correction. This can make a difference between a 10% or 95% error rate.

Response: We are now performing the multiple testing correction across all tissues jointly. The same motifs remain significant. It could have been indeed an issue if the detected motifs were very different in each tissue. We thank this reviewer to have pointed out this potential issue.

R1: Furthermore, no model selection criteria taking into account model complexity like AIC/BIC is used. As such it is not possible to assess the relevance of the new features detected.

Response: We have used cross-validation for model selection. Cross-validation yields an empirical estimation of test error, in contrast to BIC and AIC, which are theoretical estimations of test error. AIC and BIC rely on strong modeling assumptions and are therefore not preferred in practice when cross-validation is computationally cheap as here. (See also the discussion at the end of Section 7.11 of the Elements of Statistical learning.)

R1: Also, the manuscript presents through various smaller things like cut graphics in figures, und unreferenced figure panels, unclear sentences a certain lack of attention to detail, which does not increase confidence.

The Wang et al (MSB-18-8503) manuscript in contrast to the Eraslan et al (MSB-18-8513) manuscript is well written, has high quality figures & supporting figures, etc.

The Wang/8503 manuscript is publishing the generated data incl. the methods description and as such presents a classical data resource paper with a sound proteogenomics analysis but limited findings and conclusions. In contrast, the Eraslan/8513 manuscript is not only not a resource manuscript, but also presents a way less sound analysis, to the point where it's reliability is unclear, and even if true limited finding and impact.

Response: We disagree with the appreciation of the impact of this study and the reliability of the analyses, but acknowledge that there are specific concerns which we addressed with new and revised analyses and their presentation in figures and tables.

R1: Fig. 1A,B,C in MSB-18-8513 are basically equal to Fig. 1A, 2A, and 2B of MSB-18-8503, with the ones in MSB-18-8503 being generally better. Also note that for the same data 1C and MSB-18-8503 2B reported different regression slopes 2.76 and 2.6 respectively.

Response: The figures 1A, B, C now reside in MSB-18-8503. The difference between the slopes is mainly due to the FPKM > 1 filter being applied in MSB-18-8503, which is required because this manuscript focuses on PTR ratio and therefore need RNAs to be expressed in a first place. In contrast the whole unfiltered data is shown in MSB-18-8513. The resulting small difference in slopes does not affect the conclusions.

R1: Major points

1. Overall, the authors should be more carefully distinguish between correlation and causality. For example, in the abstract they refer to their work as a "catalogue of sequencing features controlling protein-to-mRNA (PTR) ratio". However, the observation that a given feature correlates with the PTR does not imply a causal relationship. More generally, let's assume A correlates with B. Let's further assume that this correlation is not random (see above for the possible FDR issue in this study). This finding could be interpreted as (i) A causes B, (ii) B causes A or (iii) A and B both correlate with the confounding factor C, which causes a spurious association between A and B. A fundamental problem throughout this paper is that only the first possibility (i.e. that a feature causes increased/decreased PTR) is considered while the third possibility (spurious correlation) is in my opinion more likely.

Response: We would like to emphasize that we have been actually cautious about this, by using systematically the word “associate” and “indicate” for all model-based evidences, by linking these associations to functional evidences when they were known, and by making clear statements that validations are required. We also undertook some functional follow-ups. As mentioned above we went through the full manuscript and identified one clearly wrong statement (a heading) which we corrected. We have furthermore found that the usage of statistical terms of “effect” and “quantification” could be misleading. Right before reporting model effects, we now stress that the effects estimated by the model may or not match effects from perturbation experiments (“We underscore that these reported effects are estimates[...]” in the result section). We have also added a new Table, Supplementary Table EV12, providing literature pointers for functional evidence if any and labeled the remaining sequence features as ‘associated’.

R1: As an example let's focus on the codon bias. In many organisms there is a significant positive correlation between gene expression level and the degree of its codon bias (for a good review see Plotkin and Kudla, Nature Reviews Genetics, 2011). Given that the PTR positively correlates with gene expression, it is expected that codon bias also correlates with the PTR. Importantly, this does not imply a causal relationship: In fact, there is "an overabundance of plausible explanatory models" for this observation (Plotkin and Kudla, 2011). The idea that "cognate tRNA concentrations influence decoding time" (page 9) is just one of many plausible explanations.

Response: Codon identity encompasses amino acid identity and synonymous codon usage. We now clarified the terminology through the manuscript. We avoid the confusing term “codon usage” and use either “codon frequency” or “synonymous codon usage” depending on what is precisely meant.

There is by now a large body of literature supporting the influence of codon identity on protein production. (Hanson and Collier, Nature Reviews Molecular Cell Biology, 2018) provides a good review of current understanding of codon usage on protein production. We have added this reference to the introduction.

In particular, the Plotkin and Kudla 2011 perturbation assay was performed in E.coli and showed no net effect on protein production of synonymous codon usage. It indicated that

translation initiation, rather than translation elongation, is the main determinant of protein production in *E. coli*. Eukaryotes have strongly different translation mechanisms. This is also in line with a more recent *E. coli* screen (Cambray et al Nat. biotech 2018). Recent experiments in yeast, supported by theoretical models, have shown that codon decoding time (and therefore both amino acid identity and synonymous codon usage) can affect the net protein production, possibly slowing down elongating ribosomes leading to traffic jams that eventually impair translation initiation (the relevant original paper (Chu et al EMBO J 2014, reviewed in Hason and Collier, Nature Reviews Molecular Cell Biology, 2018). We have added this point to the introduction.

Codon identity can also have a functional influence on PTR because amino acid identity can slow down translation elongation and because amino acid identity can affect protein stability. We are now providing a regression analysis on protein stability data that support this interpretation. We furthermore show that the regression coefficients of amino acid identity on protein stability correlate with average effects of single amino acid substitutions on protein thermodynamic stability (i.e folding), a strong determinant of protein cytoplasmic degradation (Fernando Díaz-Villanueva et al, Int J Mol Sci. 2015, PMID: 26225966).

R1: 2. Related to point 1 above, it is remarkable that almost all sequence features investigated correlated with the PTR in most of the tissues. If these features would indeed control the PTR via some mechanism, the corresponding trans-acting factors recognizing them (e.g. RNA-binding proteins, microRNAs) would have to be expressed in all tissues. I think it is more likely that observed correlations are caused by confounding factors that do not differ between tissues.

Response: A systematic technical bias would indeed show non-tissue specific associations. Association of gene region length and GC percent may in part be due to technical reasons, which is one reason why we include these as covariate. We now comment on this explicitly when we introduce the model. Furthermore, we suspect that some of our associated elements are of this kind. We emphasized this in the context of protein sequence features “Similar to the association of the 3-mer KRR, this could either reflect the effect of stretches of positively charged amino acid on slowing down translation or a technical bias due to the usage of trypsin as protein digestion enzyme (Materials and Methods, Figure 4).”

However, some core regulatory elements controlling PTR are expected to be active in every tissue. Part of this “constitutive code” includes codon identity, start and stop codon context. Supportive of our analysis to pick up genuine biological signal of this constitutive code is the fact that we recovered de novo upstream AUGs, cleavage and polyadenylation factor. There might be further constitutive pathways not known yet. Our analysis provided some new candidate elements. It is certainly conceivable that technical artefacts overload these results is possible but not necessarily “more likely” as the reviewer statement seems to imply.

The initial analysis looked at each tissue individually and thus was more likely to pick elements that explained the between-gene PTR variance rather than the per-gene between-tissue variance. Building on this and other reviewers’ comments, we are now explicitly searching for sequence elements with tissue-specific effect by performing a complementary de novo k-mer search based on gene-wise centered PTRs. This search resulted in new k-mers that are significant only in certain tissues. Figure 2 and Figure 5 were updated with the new results accordingly.

R1: 3. I am missing a systematic comparison of kmers found in this study with known RNA-binding motifs (such as in Ray et al., Nature, 2013). Such analyses might provide testable hypotheses on the molecular mechanisms regulating the PTR. Along the same lines, it might be informative to specifically look at the few RBPs (and their motifs) that are known to be expressed in a tissue-specific manner.

Response: Motif consensus sequences are now searched in the ATtRACT RNA binding protein database by using the database web interface (<https://attract.cnic.es/searchmotif>). The ATtRACT database integrates data from RBPDB, CISBP-RNA and SpliceAid-F and to our knowledge is the most extensive database containing 3,256 position weight matrices (PWMs) collected for 160 RNA binding proteins. The ATtRACT database includes the PWMs published in Ray et al Nature 2013. The RNA binding protein with the highest quality score is proposed as

possible binding candidate of the motif (new Figure EV3A and Figure 5B). We have also commented on matching tissue-specificity between k-mer effects and candidate binding proteins (e.g. QKI in brain, main text). However, as we note in the discussion, ATtTRACT matches, even with the highest scores, can be lenient. Also, this database suffers from the general poor charting of RBP binding sites. Hence, we shall take these matches with caution.

R1: 4. The authors apply a Benjamini-Hochberg multiple testing correction to the p-values when evaluating individual features. It remains however, unclear if this correction is only done for each group of feature e.g. 5' UTR sequence features, or across all features tested.

Response : The k-mer search is done by gene region individually. We do not expect and do not observe shared motifs between different regions, therefore the FDR over the different regions is likely controlled when it is controlled in each region individually. There is another significance assessment when building the full model. Here, multiple testing is done across all features of all gene regions. We also do this now over all tissues (see next question).

R1: 5. Is this testing correction done per tissue or across tissues? Since features are selected if they are below 10% error rate in at least one tissue, multiple testing correction needs to be across tissues, otherwise the true error rate will be a lot higher, namely 95% for 29 tissues!

Response: As stated above this is correct but a worst case scenario. We are now performing the multiple testing correction across all tissues jointly. All k-mers we reported previously are still significant.

R1: 6. In modelling the significance of feature and overall quality are not the only relevant criteria. The model complexity is crucial as well to demonstrate the importance of features. As such it is unclear why the author do not employ model selection criteria taking into account the number of features such as Akaike or Bayesian information criterion (AIC or BIC).

Response: Model selection criteria such as AIC and BIC measure the model goodness of fit as a balance of likelihood and model complexity. Goodness of fit is penalized as the model complexity increases in order to prevent overfitting. In our manuscript we instead used 10-fold cross validation where the model complexity is implicitly considered since R^2 values are calculated on held-out data set.

Minor points

R1: 7. Protein sequence features are mostly (i.e. with the exception of hydrophobicity) given without significance statement. Are these 10% FDR significant?

Response : The stars on Figure 4 represent $FDR < 0.1$ as mentioned in Figure 4 legend. We now moreover mention the FDR cutoff in the corresponding result text. De novo search for protein sequence motifs is also performed with $FDR < 0.1$.

R1: 8. Why was the ELM motifs assessed with the LASSO feature selection method, rather than the otherwise applied linear regression-based assessment?

Response: We have used LASSO in this step because of the large number of ELM motifs (267). LASSO could in principle also be applied to the de novo k-mer search but there we preferred to stay with the GEMMA-based approach for consistency with previous work.

R1: 9. Methods: the origin of the proteomics proteinGroups.txt is not given.

Response: The proteinGroups.txt is on PRIDE. Data resources are given in the data availability section on page 25 with login for reviewers. We now refer to the subsection Data availability at the first reference of the file "proteinGroups.txt."

R1: 10. Methods: "To estimate the mature mRNA levels, the exonic and intronic FPKM values were calculated separately and then the normalized intronic levels were subtracted from the normalized exonic levels." - From which files are FPKM values calculated and how? How were the intronic/exonic levels normalized?

Response: The two new paragraphs of the section "Protein levels, mRNA levels, and PTR ratios" in Materials and Methods describe now now this procedure in detail.

R1: 11. "Requiring at least 10 sequencing-depth normalized reads per kilobase pair further improved the correlation between mRNA and proteins, likely because of the poorer sensitivity of proteomics for lowly expressed genes." and "We set a cut-off of 10 reads per kilo base pair for a transcript to be treated as transcribed, which further improved the correlation between mRNA and proteins, likely because of the poorer sensitivity of proteomics for lowly expressed genes." - For both the mention another similarly likely reason, which should be discussed here, for improved correlation is the higher noise level of mRNA abundance calls in lower coverage RNAseq regions.

Response: We modified the sentence to include this alternative explanation.

R1: 12. "[...] the relative effects of codons in this region were highly correlating with the effects of the rest of the coding region (Materials and Methods, Spearman's $\rho=0.38$, $P<0.0024$, Fig EV5C)." - $\rho=0.38$ does not qualify as highly correlated

Response: : We re-phrased that sentence to be more accurate as "correlated significantly".

R1: 13. Fig. 1C - There are no visible data point with Brain protein levels below $1e+03$, y-axis should be the most go down to $1e+01$ not 0, or both axis should be the same. The current axis make plot look weird. Also, units (e.g. FPKM) of axis are missing.

Response: This figure panel has been now removed as it was redundant with a similar one in the other manuscript.

R1: 14. Fig. 3C - What is the UUCGCC box?

Response: We have used these two arbitrary codons to represent as a symbol for codon usage. Since we also have a panel for amino acid frequency effect now, we removed that symbol.

R1: 15. Fig. 6B - The colors of the datasets are hardly visible; e.g. reduce line thickness of mean/median line.

Response: The figure has been updated accordingly.

R1: 16. Fig. 6E,F,G,H - These panels are not referenced in the main text at all!

Response: These panels are now referenced in the corresponding result paragraphs. Also, we have expanded the text describing the assay and the results for the positive control AAUAAA (Fig 6E).

R1: 17. These graphics are not publication quality and show lack of attention to detail:

i. Fig. 1D - Sloppy x-axis labelling

ii. Fig. 2B - Icon human has lines with truncated unreadable labels.

Response: Fig 1D has been removed and we have also removed the human body icon as this was not necessary .

R1: 18. "We used as protein expression levels the per tissue median centered \log_{10} IBAQ values, settings missing values (NA) those IBAQ values equal to zero." - Please fix the language in this sentence

Response: We re-phrased this sentence of the "Protein levels, mRNA levels, and PTR ratios" Method section as well as the following sentence for consistency.

R1: 19. References are not formatted consistently nor according to journal specifications

Response : We have re-formatted them consistently. Note that MSB does not request at first submission to match the journal specifications.

Reviewer #2:

Eraslan present a in-depth analysis of sequence and experimental features of the RNA and protein data collected in the accompanying manuscript by Wang et al, quantifying expression for >11,000 genes across 29 human tissues. The work focusses on insights gained from protein-to-RNA (PTR) ratios and presents a detailed extension of work published in 2010 on one cell line.

The analysis is thorough and well-presented, but in my view, could move even more beyond the 2010 paper and make use of the new, much more extensive data available. I am excited about the work, but can only support it fully once the criticisms below have been addressed.

MAJOR

Codon optimality.

Even in the abstract the authors mention a new metric of codon optimality, but I fail to find it. The way it is written at the moment, it suggests the authors developed a new measure that helps us understand codon usage in multi-cellular organisms. I find this part of the analysis/results very interesting and important, but its presentation is entirely suboptimal.

I think the authors mean the ribosome dwell-time. That is not a new measure of codon optimality, that is just what it is: ribosome dwell-time. I would expand the discussion of what that means for translation (initiation vs. elongation etc etc), but not claim the authors developed a new measure (unless I failed to find it).

Response : This question relates to question #1 of reviewer 1. Our measure models the effect of codon identity on protein-to-RNA ratio and therefore captures not only protein production (for which codon decoding rate is one aspect) but also protein degradation. To show this, we have now performed a regression analysis of amino acid identity on protein stability. We worked at the level of amino acid because it is the common denominator between translation elongation and protein degradation mechanisms and because we and others observed that amino acid identity is a strong determinant of ribosome dwell time. Significant Spearman correlations are found between the effects on PTR ratio and ribosome dwelling time, and between the effects on PTR ratio and protein half-life but not between ribosome dwelling time and protein half-life (new Figure 3 D,F,G). Hence, our measure appears as a combination of independent effects of amino acid (and therefore codon identity) on translation elongation and on protein stability.

To give further insights and confidence that these linear regression effects on protein degradation reflects functional effects we show that these correlate well with i) hydrophobicity, and ii) with effects of single amino acid substitutions on protein thermodynamic stability (i.e folding), a strong determinant of protein cytoplasmic degradation.

The figure 3 is now focusing on codon effects only and the section has been largely rewritten.

R2: Further, it is very interesting to see this for multi-cellular organisms. Most models have analyzed single-cellular organisms. This should be looked at in more depth and/or discussed.

p. 16 middle - of course many of the codon optimality measures break down for multi-cellular organisms.

Response: The species-specific tRNA index, which estimate tRNA pool concentration from tRNA copy number in genome, could break down for multi-cellular organisms if the tRNA pools were differentially regulated across cell types. Others have indeed reported differences between proliferative and differentiating cells in human (Gingold et al, Cell, 2014). We report little tissue-specific effects, maybe because our tissues are essentially non-proliferative cell types. We include this point in discussion now.

R2: p. 16 end of second-but-last paragraph - analyzing codon optimality with PTR is NOT a new measure of codon optimality. That is entirely misleading. And also not so new. See above, dwell time is much more interesting.

Response : See above.

R2: p. 16 bottom - codon optimality does not vary much across tissues. The authors discuss this in passing, somewhat hiding a 'negative' result. Please step back from this and rework this discussion to realize that this 'negative' result is actually the interesting part: tissue-specific gene expression levels seems to be more set at the RNA than at the post-transcriptional level, as PTRs do not vary much across tissues and codon dwell time doesn't. Is that the right interpretation? This would go along with other discussions (e.g. Vogel, Marcotte Nat Rev Gen 2012 and others) on post-transcriptional regulation fine-tuning whatever has been set by RNA level... I think there is potential for some more interesting, more novel insights/discussion.

Response : This question has been well addressed by Franks et al. PLoS Comp Biol 2017. We therefore have not made it a focus of our study. We have now added an analysis of between-tissue PTR ratios and referred to Franks et al. results. In a nutshell, we agree with their result. The dynamic range of PTR ratio across genes is much larger than the dynamic range PTR ratio per gene between tissues, reflecting that the major determinant of PTRs, including codon identity, have a constitutive, non-tissue-specific effect. There are nonetheless classes of proteins that show high PTR regulation.

R2: Coupling of translation and RNA degradation.

The discussion should contain references for other work highlighting this (it exists, e.g. Sun/Cramer).

Response: We are citing four references (Hoekema et al, 1987; Presnyak et al, 2015; Mishima & Tomari, 2016; Bazzini et al, 2016). Sun,..., Cramer, Mol. Cell 2013 pointed out the coupling between mRNA degradation and transcription (not translation). Did this reviewer had another reference in mind? Please bear in mind we must be to some extent selective as we are not writing a review.

R2: Tissue-specificity.

Unfortunately, the authors miss out entirely on the enormous benefit their dataset provides: tissue-specific expression of RNA and protein. Of course the sequence information is identical, but still, there are many questions that can be addressed. And even if the data isn't good or large enough to have the resolution, data grouping or at least a discussion of trends/limitations would be novel. Otherwise, it is just an 'update' of the 2010 work. Please don't miss out on a chance to move beyond that.

Example questions would include:

- For brain and testis, with many tissue specific genes, are there trends that are different than in other tissues?
- Some of the RBP binding motifs etc that have been found, are they enriched in tissue-specific genes or in genes with particularly high/low PTRs in different tissues?
- The new/missing proteins discussed in the other paper, do they have particular characteristics?
- How does the overall model vary across tissue types? If it doesn't (low resolution), can you take the extremes, or group the data, or just compare high vs low PTR ?
- What is the significance of Fat being different in Figure 2D?
- Any tissue specific effects in Figure 4?

Response: To make a better use of the tissues-specific data while still keeping a coherence of the manuscript on its core question, we have now added a novel section "Protein-to-mRNA ratio variation of genes across tissues" and performed tissues-specific de novo search of motifs.

R2: Transcription vs. post transcriptional regulation.

Is there any way one can tie in how much is explained by just looking at RNA levels? And how much by PTR ratios (i.e. post-transcriptional regulation)? It has been done before, but a quick reference to how the new data compares would be nice.

Response: We are now addressing this in the section 'Protein-to-mRNA ratio variation across genes in each tissue' and in the section 'Protein-to-mRNA ratio variation of genes across tissues' (new Figures 1B,C). We do it using the RNA of the matched tissue and using the mRNA levels across tissues which is novel.

R2: RBP analysis.

I think this part is very interesting and also nice to see the experimental follow-up. This can be highlighted more. Also, p. 14 bottom, the authors discuss all those data but then leave the reader hanging with an interpretation or further discussion. What do these proteins do? How does this explain the trends? Do they show tissue-specific expression? Would that explain tissue-specific effects? This is a nice example to get some more discussion of tissue-specific characteristics of the data into the work, which would increase its novelty and also - by highlighting some examples in more depth, makes it much more accessible to a broader readership.

Response: The results of the RNA binding protein pull down assay show that the two positive controls recapitulate the known interacting partners. The signs of the effects we found for these

two motifs are consistent with the literature. These paragraphs mention again these pathways and their function. For the novel motif CUGUCCU, the bound proteins are in complexes (ribosome, mitochondrial ribosome, and pol(rC) complex) that are ubiquitously expressed, which is in line with the lack of tissue-specificity of the motif. Also, the CUGUCCU motif is associated with a positive effect which is in line with the ribosomal complexes and some positive effects of poly(rC) proteins on translation. Hence, because these motifs have no tissue-specificity, we cannot expand much more the interpretation of the results without being overly speculative. However, we have added an overall conclusion sentence to these results to emphasize that prediction, observed binding partners and their biological roles are consistent.

R2: The Van Nostrand data (2016) etc is still so recent that its in-depth connection to the tissue-specific data here (RBP expression, target expression etc) would be a unique chance to increase the impact of the paper. Then this should be mentioned in the abstract.

Response: The Van Nostrand 2017 data was used in the non-interpretable model but we are now providing more analyses of the Van Nostrand dataset within a new section “Tissuespecificity of RNA Binding Proteins”. In a nutshell we find that these tend to be ubiquitously expressed. This dataset did not then turn out to be as powerful as hoped to study tissue-specific regulation of PTR ratio. The Van Nostrand dataset is one of the first large scale RBP binding map and is certainly biased to the most “important” RBPs which have ubiquitous expression. Nonetheless, their protein expression levels correlate significantly with mRNA levels of their bound RNAs. We furthermore now show that these RBPs have a higher explanatory power for mRNA stability and mRNA levels (new fig EV5 E,F) than for PTR ratios.

MINOR

R2: 1. Figure 1C: it appears that low-abundance, noisy RNA has not been filtered (on the left). Perhaps at least discuss the impact of this?

Response: The former Fig 1C is now only in the companion paper (see also response to reviewer 1). All analysis of this manuscript uses a FPKM>1 cutoff.

R2: 2. I do not understand legend/figure in Figure 3C

Response: We have made the legend of Fig 3C (now Fig 3B) more explicit to explain how the coefficients of the linear model relates to the effect of a two-fold change in codon frequency on PTR ratio (y-axis). We also explain the ordering of the boxes. Furthermore we have added a new figure, Figure 3A that shows the amino acid frequency effects.

Reviewer #3:

The manuscript "Quantification and discovery of sequence determinants of protein per mRNA amount in 29 human tissues" by Eraslan et al uses regression analysis to study the impact of various sequence features on protein-to-mRNA (PTR) ratios. This is essentially a follow-up on an earlier report in MSB (Vogel et al, 2010), but the improvement in data size and quality is substantial enough to warrant a repetition of the pioneering study. In addition, to my knowledge Eraslan et al are the first to use such data to search for - and identify - novel sequence motifs that may control PTR ratios. As such I believe this manuscript is a valuable addition to our current understanding of mRNA and protein expression control and provides a blueprint for similar association studies in the future.

There are a few points that need to be addressed:

Overlap with accompanying paper: Although the manuscripts by Wang et al and Eraslan et al can stand on their own, there is at present still some unnecessary overlap. The conclusion that there is a quadratic relationship between mRNA and protein levels in every tissue should not be presented in both papers, particularly because it only confirms earlier reports by the authors and other groups. For example, Fig. 1C in Eraslan et al is almost identical to Fig. 2B in Wang et al. The authors need to decide which of the two reports will include that finding, and the other paper should then refer to this rather than showing nearly identical material.

Response: We decided to have the figures 1A, B, C in MSB-18-8503 only and refer to it in this manuscript.

R3: 1 Transcript isoforms: The authors select the major transcript isoform for each gene based on its average protein abundance across all tissues. I think they need to justify that decision and provide some statistics to show if that is a sensible approach. For example, are the most abundant protein isoforms generally produced by the most abundant transcripts? Are there any large transcript isoform differences across the tissues, i.e. is it reasonable to use one transcript isoform per gene across all tissues?

Response: We have now quantified mRNA isoform expression using the software Kallisto (Bray et al, 2016). This analysis reports that nearly half (48%) the genes has a single major isoform across all tissues at the mRNA level. At the protein level, the number of genes with multiple quantified isoforms at the protein level was small (10% of the detected genes). There are technical and possibly biological reasons for identifying less isoforms at the proteins than at the mRNA level. However, both levels indicate that using a single isoform is a reasonable assumption.

Moreover, the major transcript isoform matches the major protein major isoform in every tissue they were measured for 4,250 genes (36%). Also, in most tissues, the protein major isoform is also the major mRNA isoform for the majority of the genes (new Appendix Fig S2). These results are now reported in the section “Matched transcriptomic and proteomic analysis of 29 human tissues”.

R3: 1 Dynamic range: The dynamic range of proteins shown in this paper is 1-2 orders of magnitude lower than in the accompanying paper. Why is that? Perhaps because the subset of proteins included here are enriched for housekeeping functions? It would be good to show that, because gene regulation of constitutively expressed and developmentally induced (tissue specific) genes can be quite different. This would have implications for their overall conclusions.

Response: The data is the same. However, we are reporting in this manuscript 80% equi-tailed intervals which provide very robust but also more conservative measures of dynamic ranges. To avoid confusion, we are now only referring to these intervals as 80% equi-tailed interval and do not coin these “dynamic ranges”.

R3: In Fig 1B, I also find it confusing to show two completely different units (FPKMs and iBAQs) with the same x axis scale.

Response: This figure resides only in the accompanying manuscript now.

R3: 1 Perhaps I missed this, but it would be interesting to know how much protein-to-mRNA ratios actually vary between tissues. I suppose there are some genes where the ratio varies more and others that vary less, so that may be interesting to point out.

Response: In the new ‘Protein-to-mRNA ratio variation of genes across tissues’ section, we are now providing relevant statistics and gene set enrichment analysis on across-tissue PTR ratio variance. This question has been well investigated by Franks et al. 2017. We therefore do not expand much on this.

R3: 1 Out-of-frame AUGs in the 5' UTR decrease protein-to-mRNA ratios. This is an interesting observation. Producing a non-functional or unstable protein seems like quite a wasteful way of reducing a gene's PTR, so I'm wondering how many out-of-frame AUGs we are talking about here?

Response: As stated in the manuscript text and displayed in Figure EV4A, 5,093 genes out of 11,575 inspected genes has at least 1 out-frame uAUG. Upstream ORFs have been well studied and are known to be widespread in the human genome (Kozak et al, NAR, 1987; Kochetov et al Mol. Genet. Genomics . 2005). See also Patrick McGillivray et al, NAR 2018 for a recent study.

Are out-of-frame AUGs in general depleted from 5' UTRs, as one might expect? In other words, are these cases where gene expression has yet to be optimised more completely in evolutionary terms? Or could it be that such out-of-frame translation gives rise to

stable but unannotated gene products?

Response: This is unclear. uAUG are generally depleted in 5'UTR. See Churbanov et al. NAR 2005 (Table 8) for human, mouse, rat, and yeast. "Optimality" of gene expression does not necessarily mean energetically optimal. Actually the uORFs are integral parts of protein expression regulation and many may be under positive selection pressure. In line with that, we now explain in the "mRNA 5' UTR sequence features" section that these uAUGs are evolutionarily conserved and display the mean conservation scores at the motif and 10 nucleotide flanking sides in Figure EV2C. We also refer to earlier results by Churbanov et al. NAR 2005. Finally, the number of uORFs may be underestimated and mass-spectrometry could help discovering some. We have some candidate examples in the companion paper.

R3: I The novel 5 k-mers: With an FDR of 10% there is a good chance that one of the 5 motifs is a false positive, so I think more data are needed to allow the reader to get a better idea of how likely each of these motifs are to have an effect. For example, the motif CUGUCCU is found in 323 genes. When you say it increases PTR by 33%, is that supported by all these genes or are there perhaps a few genes with this motif that have a dramatically increased PTR, thus driving the average impact of the motif?

Response: To the first part of the question: We report now more motifs because we have done a tissue-specific k-mer search. We are now highlighting the conservation analysis, which shows that 12 of 25 5'UTR kmers (including de novo found uAUG) and 11 out of 20 3'UTR kmers (including the 4 known ones) are more conserved than their flanking regions. We have also performed matches to the ATtRACT database, as indication of possible RBP recognizing these motifs. ATtRACT matches, even with the highest scores, can be lenient. Also, this database suffers from the general poor charting of RBP binding sites. Hence, we shall take these matches as indicative and with caution.

To the second part of the question: Our estimations are based on a linear model. Hence they represent average effects, assuming additive of the effects (on the logarithm scale). In reality, the elements likely work non-additively with effects that depend on the sequence context, and position along the mRNAs. Hence, we can only predict an expected effect on given gene. With this amount of data, modeling interaction terms is practically not possible.

A new paragraph in the discussion (2nd paragraph) underscores these limitations.

R3: Likewise, the fact that these genes are enriched for certain GO terms really doesn't mean much without reporting an actual fold-enrichment, or percent of genes associated with those GO terms and p-values etc.

Response: In Appendix Figures S6 and S12 we now provide the enrichment of GO terms with their FDR values.

R3: Finally, the evolutionary conservation is quite critical here. It looks visually convincing for three motifs but not for CUGUCCU and GGCGCCCG. It would be good to have some statistical test (rather than just the sequence plot) to support the fact that these motifs are evolutionarily conserved in a statistically significant manner.

Response: In phylogenetic conservation plots (Appendix Figures S5 and S11) we now provide the p-values of the Wilcoxon one-sided rank-sum test performed to evaluate the significance of the enrichment at the motif site compared to [-10,+10] nucleotide flanking regions.

R3: I Codon frequency vs amino-acid frequency: The authors claim that codon frequency is the single most important sequence feature to predict PTR ratios, but I'm not sure their data fully support this conclusion. In Fig 3C it appears that the PTR ratios depend mainly on the amino acid frequency, as most codons that encode for the same amino acid show the same directionality of PTR changes. Moreover, the authors state the predictive power of codon frequency is only 2% higher than that of amino acid frequency (17% vs 15%), which seems to me would support the conclusion that the latter is more important, since codon frequency will reflect amino acid frequency more strongly than the other way around. Perhaps the authors can dissect this issue a bit further. For example, how important are codon vs aa frequency when you distinguish between amino acids encoded by multiple codons vs single codons? Can you find a way to better distinguish between these two effects? I think this is quite important since the biological mechanisms behind them will be quite different - codon usage points to an impact of the gene expression programme (availability of tRNAs etc), while aa frequency points to

biological properties of the proteins themselves (protein stability or function). Both effects will probably play some role but the relative contribution of each would be important to estimate. Especially because codon usage (or aa frequency) is by far the most important feature for PTR prediction in the multivariate regression.

Response: This is correct. We have re-worked Figure 3 and the corresponding text to clarify the contribution of amino acid identity versus synonymous codon usage. We also now analyse protein-half-life and ribosome dwelling time at these two levels. Furthermore, we report explained variances at these two levels separately (Figures EV5C-F), even though in our model we use the log2 frequencies of codons as covariate.

R3: I De novo motifs when controlling for codon pair frequencies (last paragraph of "Codon usage" section): Why were these motifs not picked up in the earlier motif search?

Response: The earlier motif search is for UTR. We also find these 6-mers if we do not correct for codon pair frequencies.

R3: Given that there is no evidence for these motifs to be conserved and that they are found in rather selective frames / positions, I would say this is too speculative to be included without additional support.

Response: We excluded these two motifs from our model and the analysis now.

R3: I Protein sequence features: I think there is a danger here to confuse cause with correlation. For example, just because nuclear localisation signals correlate with lower PTR ratios does not mean that the signal peptide sequence itself is mechanistically involved in it. Isn't it more likely that nuclear proteins per se might have lower PTR ratios? That could simply be driven by a higher proteasome activity in the nucleus. For example, I believe proteins involved in nuclear RNA pre-processing are generally less stable than cytoplasmic proteins. In other words, what if the sequence motifs you pick up are simply an indirect measure of protein function? I guess it's impossible to really address that without biological follow-ups (as you perform for one motif later), but maybe you could address this point by associating GO terms with PTR ratios. If nuclear proteins have a lower PTR ratio in general, then it is unlikely that the NLS contributes directly to it. I understand the goal is to predict PTR ratios from gene sequence, but while some of the features you describe are directly related to the mechanisms of translation / degradation, the impact of others may be very indirect.

Response: We agree that without validation experiments these associations cannot be interpreted as causal. Nevertheless, we performed the analysis the reviewer suggested and compared PTR ratios of nuclear (GO:0005634) vs. other proteins and additionally we applied the same comparison on average protein half-lives of five cell types obtained from publicly available data sets. As we now explain in "Protein Sequence Features" section and show in Appendix Figure S9, there was no significant PTR ratio difference between nuclear (GO:0005634) and non-nuclear proteins. Moreover, we now show that this motifs also associate with average protein half-lives. These 6 linear protein motifs were significantly associated with shorter protein half-lives (new Appendix Figure S10), and nuclear proteins with the four nuclear localization signals were associated with even shorter half-lives compared to nuclear proteins without these signals.

R3: I Variance explained across tissues: With the multivariate regression models, where you combine the individual features to predict PTRs, I think you need to be careful with the phrasing. The phrase "This model explained 22% of the variance in median across tissues" to me suggests that you explained the difference in PTRs of one gene across many tissues, whereas I believe you are explaining the variance of PTRs between genes, showing the distribution of those predictions across the 29 tissues as boxplots. This should be formulated a bit more carefully to avoid confusion.

Response: We re-phrased this as: "This model explained 22% (median across tissues) of the variance".

R3: I Given the central importance that the authors attribute to the value 3.2 this deserves a representation in a figure. The currently closest panel is Fig 5B. A light reader could draw from this the impression that the model predicts a PTR ratio to be about 1e+05.

Response: The former fig 5B had a very wide x-axis (matching the y-axis range) which led

indeed to the impression to light reader that the predictions were narrow around $1e+05$. We have now updated the x-axis range to the data.

R3: 1 The conclusion that "The predicted PTR ratios positively correlated with the mRNA levels (Fig 5D)." should probably not be drawn without a reference to the very broad spread also visible in that figure.

Response: We have changed this sentence to "Moreover, we observed that the predicted PTR ratios moderately positively correlated with the mRNA levels (Figure 5D, Spearman's $\rho=0.26$)"

R3: 1 The speculation "We suggest that the uncoupling of transcription and translation underlies a fundamental difference in the relationship between protein and mRNA levels across genes in eukaryotes compared to prokaryotes and may allow protein copy numbers of eukaryotic cells to span a much larger dynamic range." falls short in recognising evolutionary considerations as voiced by the Hurst lab, for example, and backed in vertebrates by recent work of the Rappsilber lab.

Response: We are not sure what exact publications of Rappsilber and of Hurst was meant here. The Rappsilber lab has published recent studies in *Mol Syst Biol* and in *Mol & Cell Proteomics* that showed that non-functional, mRNA co-expression of neighboring genes in the human genome is buffered at the protein levels, which is related to Hurst 2004 nature genetics article on eukaryotic genome organization. This is however remotely related to our point on prokaryotic versus eukaryotic expression. A more specific suggestion would help us.

R3: 1 Figure legends could help readers better in understanding the displayed data, for example by spelling out acronyms (e.g. `LIG_KEPE_1` and others used in Figure 4 are not intuitively understood by me).

Response: In Figure 4 the eukaryotic linear protein motif acronyms are now explained in the legend text. Also, other figure legends have been updated to be more explicit.

2nd Editorial Decision

8 January 2019

Thank you for sending us your revised manuscript. We have now heard back from the two reviewers who were asked to evaluate your study. As you will see below, the reviewers think that the work has benefited from the performed revisions and the additional experimental analyses. They are however still not convinced that the study is suitable for publication. Both reviewers raise several concerns, most of which are related to presentation issues. We would therefore ask you to perform another revision and to carefully re-write the manuscript in order to address these remaining issues and to make the findings easily accessible to the reader.

REFeree REPORT

Reviewer #2:

Eraslan present a revised manuscript in which they investigate the ability of sequence and experimental information to explain mRNA, protein concentrations, protein-to-mRNA ratios (PTR), and half-lives across the 29 tissue measurements. They present a multivariate model which explains 22% of the variation in PTR ratios across genes (and tissues) based on sequence features alone and 27% of the variation if other data is included.

The work is thorough and has interesting results and it benefitted much from addressing the last comments. In particular, the authors strictly focus on associations (and correlations) rather than implying causation. The work presents an important complementary analysis to the other paper. It has several new findings, such as the dominant effect of sequence features (explaining 22% of the variance) which comprises most of what post-transcriptional data explains (27%), and the binding of some RBPs, in addition to the dominant role of RBPs over miRNAs in controlling the fate of mRNAs (that is new!) etc. However, the work is very difficult to read which hinders an outsider to grasp its scope.

Specifically, my remaining criticisms are:

1. The writing can still be improved. The manuscript has plenty of typos and language glitches, I strongly recommend proof-reading by a native speaker.
2. I still fail to find a clear, direct definition of the "new metric of codon optimality". I could not see it in the Methods, nor Introduction or Results. It receives a name in the Discussion (PTRAI), but I cannot find a single quantitative description (i.e. formula). I suspect it is the "codon effect on PTR" - but even that is not clearly explained what the authors mean. The manuscript would benefit from clear definition and subsequent use of one and only one phrase.
3. Similar arguments apply to 'codon dwell time' which seems also be called decoding time, ribosome dwell time, codon dwelling time etc. Again, I am missing a clear definition and consistent use of one phrase. Even if the measure has been taken from a different publication, it is not common enough yet to warrant skipping an explanation.
4. The authors use "effect of" excessively and perhaps sometimes unnecessarily. Correlations between the "effect of PTR ratios" might just be the correlation between PTR ratios and something else.
5. Since RNA concentrations, protein concentrations, PTR ratios and RNA and protein half-lives are all inter-correlated, it is important to clarify throughout the manuscript which other correlations (or associations) were detected _accounting for_ partial correlations. I.e. if the "new metric of codon optimality" correlates with PTR ratio and protein half-life, is that independent of a correlation between PTR ratio and protein half-life? The authors clarified this for ribosome dwell time, but not for other contexts.
6. If I understand correctly, the final model can explain a median of 22% in variation of PTR ratios purely with sequence features and explains 27% if experimental measures are included. I find that interesting and new - that so little of the new datasets of post-transcriptional regulation seems to be encoded in sequence.
7. The authors mention several times that "our integrative model predicts PTR ratios at a median precision of 3.2-fold." - I find it confusing to measure precision in 'fold' changes - what does that mean?
8. Minor: the authors clearly move beyond Vogel et al.'s work from 2010 - no need to justify their own work twice with a whole paragraph.
9. Minor: There are several awkward phrases, like "variance in protein log-transformed fold-changes". Please edit.
10. What are "2-fold amino acid frequency increase effects" in Figure 3? The phrase is unclear.
11. Minor: The manuscript would benefit from adding simple, plain language summaries to each paragraph to increase readability. E.g. phrases like "associated with a lower PTR ratio between 4 nt 5' of the start codon and 4 nt 3' of the start codon" are hard to digest and could be paraphrased by something like "We find low protein-to-mRNA ratios immediately surrounding start codons". Another example of difficult language is "In contrast, negative minimum folding energy in 51 nt windows associated positively with the PTR ratio between 41 nt and 55 nt 3' of the start codon"
12. I find the "out-of-frame" phrase confusing and do not know what the authors mean. Out-of-frame - do they mean outside the main ORF, i.e. upstream of the main ORF in the 5'UTR, or do they mean out-of-frame, i.e. shifted by one or two nucleotides compared to the frame of the main ORF? Please define and stick to one and only one phrase. E.g. What does this mean? "a single out-of-frame uAUG is associated with a 20% reduced PTR ratio compared to a single out-of-frame uORF"
13. Last paragraph on page 12: "We then asked whether our estimated codon effects on PTR ratios also captured effects of amino acids on protein degradation." - First, I do not know what codon effects on PTR ratio are precisely. Second, the paragraph does not comment on that but on correlations with amino acid types (or frequencies???) .

Reviewer #3:

I agree with reviewer 1 that there is a general sloppiness and lack of attention to detail both in the writing and the figure preparation that undermines the credibility of the manuscript and that falls short of the quality I would expect for a paper in MSB. I have the impression that this revised version is harder to read and follow than the earlier one. This is a shame as clearly a lot of work went into the actual experiments and the results per se are quite interesting. To give you a few examples (not an exhaustive list by any means):

I have trouble following the logic of the paragraph entitled "Protein-to-mRNA ratio variation of genes across tissues". First you explain that between 0 and 43% of the variance in protein fold-changes can be explained by the corresponding mRNA fold-changes. That makes sense and is perfectly consistent with values reported by others. But then you say between 7 and 51% of the variance of protein fold-changes is explained by "mRNA levels across tissues". How is that different? Moreover, in the corresponding figure axis "mRNA levels across tissues" is replaced by "mRNA fold-changes of all tissues". That's a completely different thing. And how do measure "mRNA fold-changes of all tissues"? Do you mean the mean fold-change? Is it log2 or log10 transformed?

Finally, the same sentence continues with "which further indicates that co-expression patterns of mRNAs may be predictive of posttranscriptional regulation". How does co-expression come into this picture suddenly?

In the introduction: "Decades of single gene studies have revealed numerous sequence features determining protein-per-mRNA copy numbers affecting initiation, elongation, and termination of translation as well as protein degradation." What are "protein-to-mRNA copy numbers"? And it sounds like the copy numbers affect initiation etc...

The sentence "5,636 (43%) genes had a single major isoform across all tissues". I presume you meant to say that for these 5,636 genes the same isoform was the most abundant one across all tissues?

In the appendix, the supplementary figures have no legends or legends that are much too short and uninformative (basically just titles).

Lack of clarity in figure 1B, C: Why use different scales for the x- and y-axes and between B and C? It should be 0 to 0.6 on all four axes, otherwise you can't compare the plots properly. There is also an unexplained line which looks like a misplaced regression line (it's actually a $y \sim x$ curve) and it has a different colour in the two panels.

2nd Revision - authors' response

22 January 2019

Reviewer #2:

Eraslan present a revised manuscript in which they investigate the ability of sequence and experimental information to explain mRNA, protein concentrations, protein-to-mRNA ratios (PTR), and half-lives across the 29 tissue measurements. They present a multivariate model which explains 22% of the variation in PTR ratios across genes (and tissues) based on sequence features alone and 27% of the variation if other data is included.

The work is thorough and has interesting results and it benefitted much from addressing the last comments. In particular, the authors strictly focus on associations (and correlations) rather than implying causation. The work presents an important complementary analysis to the other paper. It has several new findings, such as the dominant effect of sequence features (explaining 22% of the variance) which comprises most of what post-transcriptional data explains (27%), and the binding of some RBPs, in addition to the dominant role of RBPs over miRNAs in controlling the fate of mRNAs (that is new!) etc. However, the work is very difficult to read which hinders an outsider to grasp its scope.

Specifically, my remaining criticisms are:

R2: 1. The writing can still be improved. The manuscript has plenty of typos and language glitches, I strongly recommend proof-reading by a native speaker.

Response: We apologize for the remaining typos and unclear usage of language. We carefully and thoroughly edited the manuscript, including professional proof-reading/editing by a native speaker.

R2: 2. I still fail to find a clear, direct definition of the "new metric of codon optimality". I could not see it in the Methods, nor Introduction or Results. It receives a name in the Discussion (PTRAI), but I cannot find a single quantitative description (i.e. formula). I suspect it is the "codon effect on PTR" - but even that is not clearly explained what the authors mean. The manuscript would benefit from clear definition and subsequent use of one and only one phrase.

Response: We agree with the reviewer comment. The reviewer is correct: PTR-AI is the same as the codon effect on PTR. We now define PTR-AI in the paragraph relating to Figure 3 in plain words as the protein-to-mRNA ratio fold-change associated with doubling the frequency of a codon. From there on, we use PTR-AI systematically instead of codon effect on PTR. We also added in Materials and Methods a PTR-AI subsection describing how PTR-AI is computed from the fitted coefficients of the models.

R2: 3. Similar arguments apply to 'codon dwell time' which seems also be called decoding time, ribosome dwell time, codon dwelling time etc. Again, I am missing a clear definition and consistent use of one phrase. Even if the measure has been taken from a different publication, it is not common enough yet to warrant skipping an explanation.

Response: We now use codon decoding time systematically, defined as the typical time it takes a ribosome to decode a codon according to Dana and Tuller, NAR, 2014.

R2: 4. The authors use "effect of" excessively and perhaps sometimes unnecessarily. Correlations between the "effect of PTR ratios" might just be the correlation between PTR ratios and something else.

Response: We systematically revised every sentence using the word "effect". We found instances in the vein of what the reviewer points out such as a feature "associated with the largest effect" which can be replaced by "showed the strongest association" (e.g. Kozak). While this can be simplified, other instances cannot. For instance, there is a correlation between PTR ratio and protein half-life.

Nonetheless, we find it important to show that the effects of amino acid on PTR ratio correlate with the effects of amino acid on protein half-life (Fig 3F), indicating that the role of amino acid frequency on protein stability drives their association with the PTR ratio. In contrast, the stop codon context e.g. has a strong effect on PTR ratio but not on protein half-life.

Also, we have removed instances that were not needed. These include mechanistic effects that can sometimes be described as "role" and unnecessary usages of the word effect (e.g. "this apparent amplification effect" -> "this apparent amplification"). However, for consistency, we keep using the term effect throughout the manuscript to describe the effects estimated by the model.

R2: 5. Since RNA concentrations, protein concentrations, PTR ratios and RNA and protein half-lives are all inter-correlated, it is important to clarify throughout the manuscript which other correlations (or associations) were detected _accounting for_ partial correlations. I.e. if the "new metric of codon optimality" correlates with PTR ratio and protein half-life, is that independent of a correlation between PTR ratio and protein half-life? The authors clarified this for ribosome dwell time, but not for other contexts.

Response: We are not accounting for partial correlations among any of RNA concentrations, protein concentrations, PTR ratios and RNA and protein half-lives. Hence, the "new metric of codon optimality" (PTR-AI) correlates with the effect of codon frequency on protein half-life, not accounting for the correlation between protein half-life and PTR ratio. This does not affect our conclusions as it is consistent with the model: codon frequency -> amino acid frequency -> protein stability -> PTR ratio. The similar analysis with ribosome dwell time (now codon decoding time) the reviewer refers to supports the second source of contribution of codon on PTR ratio: codon frequency -> translation -> PTR ratio.

Since correlations rather than partial correlations are the default, we did not want to clutter the text further by explicitly stating when correlations are not partial. However, we would like to emphasize that our reported associations between every sequence feature and PTR ratio are accounting for the other sequence features as well. We made this clear when introducing the model with statement: "The effects of each sequence feature on PTR log-ratio are estimated using the joint model, thereby controlling for the additive contribution of all other sequence features (Table EV7)."

R2: 6. If I understand correctly, the final model can explain a median of 22% in variation of PTR ratios purely with sequence features and explains 27% if experimental measures are included. I find that interesting and new - that so little of the new datasets of post-transcriptional regulation seems to be encoded in sequence.

Response: Indeed, using experimentally measured binding of RBPs, miRNA binding sites, m6A modification, protein post-translational modifications increased substantially the explained variance over sequence only. However, this does not imply that these post-transcriptional events are not driven by regulatory elements encoded in sequence. Rather, this may reflect that our regression of PTR ratio on sequence was not powerful enough to capture those potentially complex regulatory sequence elements. In fact, for exactly this reason, we used the direct experimental evidences of these post-transcriptional events because they were difficult to capture from a regression of PTR ratio on sequence. We now mention this point in the text after reporting the 27% explained variance.

R2: 7. The authors mention several times that "our integrative model predicts PTR ratios at a median precision of 3.2-fold." - I find it confusing to measure precision in 'fold' changes - what does that mean?

Response: We work on the log10 scale. The median error, i.e. the median of $\log_{10}(\text{measured PTR}) - \log_{10}(\text{predicted PTR})$, equals $\log_{10}(3.2)$. Thus, a prediction is, in median, 3.2-fold away from the measurement.

R2: 8. Minor: the authors clearly move beyond Vogel et al.'s work from 2010 - no need to justify their own work twice with a whole paragraph.

Response: We agree with the reviewer. The two paragraphs were indeed redundant. We removed the discussion paragraph.

R2: 9. Minor: There are several awkward phrases, like "variance in protein log-transformed foldchanges". Please edit.

Response: We have edited the text to remove awkward phrases including this one, also with the aid of a professional proof-reading service.

R2: 10. What are "2-fold amino acid frequency increase effects" in Figure 3? The phrase is unclear.

Response: We agree that this is one of the instances which is difficult to read. To make clear what this means, we now re-labeled Fig 3A with a panel title "Amino acid frequency increase (2-fold)" and a y-axis title "Effect on PTR-ratio". We similarly modified the other panels of Fig 3. Also in the text, we define the "amino acid effect" as the fold-change (of PTR-ratio or of protein half-life depending on the situation) associated with doubling the frequency of an amino acid.

R2: 11. Minor: The manuscript would benefit from adding simple, plain language summaries to each paragraph to increase readability. E.g. phrases like "associated with a lower PTR ratio between 4 nt 5' of the start codon and 4 nt 3' of the start codon" are hard to digest and could be paraphrased by something like "We find low protein-to-mRNA ratios immediately surrounding start codons". Another example of difficult language is "In contrast, negative minimum folding energy in 51 nt windows associated positively with the PTR ratio between 41 nt and 55 nt 3' of the start codon"

Response: We agree with the reviewer and have revised the text to improve its readability, including the expressions mentioned by the reviewer.

R2: 12. I find the "out-of-frame" phrase confusing and do not know what the authors mean. Out-of-frame

- do they mean outside the main ORF, i.e. upstream of the main ORF in the 5'UTR, or do they mean out-of-frame, i.e. shifted by one or two nucleotides compared to the frame of the main ORF? Please define and stick to one and only one phrase. E.g. What does this mean? "a single out-of-frame uAUG is associated with a 20% reduced PTR ratio compared to a single out-of-frame uORF"

Response: We mean shifted by one or two nucleotides compared to the frame of the main ORF. Because we consider upstream AUG (uAUG) and upstream ORFs (uORFs), which are outside the coding frame in the first place, the term is not ambiguous. It is also frequently used in this sense in the literature (e.g. Dvir et al PNAS 2013). We now specify "out-of-frame relative to the main ORF" at the first occurrence.

13. Last paragraph on page 12: "We then asked whether our estimated codon effects on PTR ratios also captured effects of amino acids on protein degradation." - First, I do not know what codon effects on PTR ratio are precisely. Second, the paragraph does not comment on that but on correlations with amino acid types (or frequencies???)

Response: We agree with the reviewer that this expression is not precise. To the first point: As mentioned above, we now define PTR-AI (former "codon effect on PTR ratios"). We also now define the amino acid effect on PTR ratio as the PTR ratio fold-change associated with doubling the frequency of an amino acid. To the second point: We have edited that sentence and the entire paragraph. We now report two distinct analyses: one at the level of amino acid frequencies, and one at the level of codon frequencies. Both give the same message.

Reviewer #3:

I agree with reviewer 1 that there is a general sloppiness and lack of attention to detail both in the writing and the figure preparation that undermines the credibility of the manuscript and that falls short of the quality I would expect for a paper in MSB. I have the impression that this revised version is harder to read and follow than the earlier one. This is a shame as clearly a lot of work went into the actual experiments and the results per se are quite interesting.

Response: We have edited the text to improve its readability and intelligibility and to resolve ambiguities (see response to reviewer 2).

R3: To give you a few examples (not an exhaustive list by any means):

I have trouble following the logic of the paragraph entitled "Protein-to-mRNA ratio variation of genes across tissues". First you explain that between 0 and 43% of the variance in protein fold-changes can be explained by the corresponding mRNA fold-changes. That makes sense and is perfectly consistent with values reported by others. But then you say between 7 and 51% of the variance of protein foldchanges

is explained by "mRNA levels across tissues". How is that different?

Response: We agree with the reviewer that the logic is difficult to follow, and have rewritten this

paragraph. We now define the relative protein level as the log-ratio of the protein level compared to its median level across tissues. We similarly define relative mRNA levels. In the first scenario mentioned by the reviewer, we ask how well one can predict the relative protein level of a gene in one tissue given its relative mRNA level in the same tissue. In the second scenario, we ask how well we can predict the relative protein level of a gene in one tissue given its relative mRNA levels in each of the 29 tissues (in other words, the mRNA expression profile across 29 tissues). The predictions improve when using the relative mRNA levels from all tissues because it benefits from more information.

R3: Moreover, in the corresponding figure axis "mRNA levels across tissues" is replaced by "mRNA fold-changes of all tissues". That's a completely different thing. And how do measure "mRNA foldchanges

of all tissues"? Do you mean the mean fold-change? Is it log2 or log10 transformed?

Response: The figure axis was correct. We now use the term 'relative level' consistently, as defined above. We use the log10 transformation. However, the choice of the base of the logarithm has no effect on the explained variances as long as the same base is used for mRNA and for protein. We therefore do not specify it.

R3: Finally, the same sentence continues with "which further indicates that co-expression patterns of mRNAs may be predictive of posttranscriptional regulation". How does co-expression come into this picture suddenly?

Response: We agree with the reviewer that this was not entirely clear from the previous version of the text. To briefly elaborate on this: Having relative mRNA levels for all tissues and not only for the tissue of interest contributes mRNA co-expression information. Biologically, it is conceivable that co-expression patterns of mRNAs can be predictive for post-transcriptional regulation because genes of the same pathways are often also co-regulated at the mRNA level and because of coordinated post-transcriptional regulation within pathways (Franks et al. 2017). We now make this interpretation clear in the text. We introduce this already in the section "Protein-to-mRNA ratio variation across genes in each tissue" because the argument is the same.

R3: In the introduction: "Decades of single gene studies have revealed numerous sequence features determining protein-per-mRNA copy numbers affecting initiation, elongation, and termination of translation as well as protein degradation." What are "protein-to-mRNA copy numbers"? And it sounds like the copy numbers affect initiation etc...

Response: We apologize for the confusion. We used the term "copy numbers" in the sense of "number of molecule copies per cell" as often done in quantitative biology (simply a different term for protein or mRNA concentration). To avoid further confusion, we re-phrased the term with "the number of protein molecules per mRNA molecule".

R3: The sentence "5,636 (43%) genes had a single major isoform across all tissues". I presume you meant to say that for these 5,636 genes the same isoform was the most abundant one across all tissues?

Response: Thank you for pointing this out. We rephrased that sentence.

R3: In the appendix, the supplementary figures have no legends or legends that are much too short and uninformative (basically just titles).

Response: We have revised the supplementary figure legends.

Lack of clarity in figure 1B, C: Why use different scales for the x- and y-axes and between B and C? It should be 0 to 0.6 on all four axes, otherwise you can't compare the plots properly. There is also an unexplained line which looks like a misplaced regression line (it's actually a $y \sim x$ curve) and it has a different colour in the two panels.

Response: We keep the scale so that labels remain legible. However, we have added a label " $y=x$ " and now use the same colour for both panels.

Corresponding Author Name: Julien Gagneur

Manuscript Number: MSB-18-8513